# Climatologies and long-term changes of mesospheric wind and wave measurements based on radar observations at high and mid-latitudes.

Sven Wilhelm[1], Gunter Stober[1,3,4], and Peter Brown[2]

[1]Leibniz Institute of Atmospheric Physics at the University of Rostock, Kühlungsborn, Germany

[2]Department of Physics and Astronomy, Western University, London, Ontario, Canada

[3]Institute of Applied Physics, University of Bern, Bern, Switzerland

[4]Oeschger Centre for Climate Change Research, University of Bern, Switzerland

**Correspondence:** S. Wilhelm (wilhelm@iap-kborn.de)

**Abstract.** We report on long-term observations of atmospheric parameters in the mesosphere and lower thermosphere (MLT) made over the last two decades. Within this study, we show based on meteor wind measurement, the long-term variability of winds, tides, and kinetic energy of planetary and gravity waves. These measurements were done between the years 2002 and 2018 for the high latitude location of Andenes (69.3°N, 16°E) and the mid-latitude locations of Juliusruh (54.6°N, 13.4°E) and Tavistock (43.3°N, 80.8°W). While the climatologies for each location show a similar pattern, the locations differ strongly with respect to altitude and season of several parameters. Our results show annual wind tendencies for Andenes which are toward the south and to the west, with changes of up to 3 $ms^{-1}$ per decade, while the mid-latitude locations showing smaller opposite tendencies to negligible changes. The diurnal tides show nearly no significant long-term changes, while changes for the semidiurnal tides differ regarding altitude. Andenes shows only during winter a tidal weakening above 90 km, while for CMOR occur an enhancement of the semidiurnal tides during the winter and a weakening during fall. Furthermore, the kinetic energy for planetary waves showed strong peak values during winters which also featured the occurrence of sudden stratospheric warming. The influence of the 11-year solar cycle on the winds and tides is presented. The amplitudes of the mean winds exhibit a significant amplitude response, for the zonal component below 82 km during summer and from November - December between 84 and 95 km at Andenes and CMOR. The SDT show a clear 11-year response at all locations, from October to November.

# 1 Introduction

Over the last several decades, studies of wind and wave action in the mesosphere and lower thermosphere (MLT) have focused on coupling processes to layers above and below (e.g., Yiğit et al., 2016), dynamical processes of the wind (e.g., Fritts and Alexander, 2003), the local variability of the measured winds (e.g., Stober et al., 2018), and long-term changes (LTC) of winds and waves (e.g., Keuer et al., 2007). Wind measurements at these heights rely mainly on remote sensing techniques, like satellites, lidars, radars, and passive mircowave radiometer. Each of these techniques have its own strengths and limitations with regard to the time and altitude resolution or measurement conditions. Meteor radar wind observations of the MLT have a long proven record due to their reliable, long-term measurement capability, independent of weather conditions. These radars detect the ionized plasma trails of meteors left behind after the hypersonic passage of meteoroids in the Earth's atmosphere. The resulting meteor trails drift with the neutral background wind. By measuring the radial velocities and the positions of the trail echoes in the sky, wind velocities of the atmosphere can be determined. The measurements of these local winds and the associated tides are key inputs to validate and update global circulation models. Basically, climatologies of winds and tides in the mesosphere are only partly well-represented in GCMs. In particular at the MLT differences occur between models and observations, which are outlined in several studies. Yuan et al. (2008a) showed differences between some models and observations, as well as, also between models itself, by mentioning that the height of the summer mesopause differs. Stronger differences occur during the winter, opposite prevailing wind directions occur above the mesopause between models and observations (e.g., Pokhotelov et al., 2018). A reason for these differences is probably based in using different gravity waves parameterizations.

The general circulation of the MLT is strongly influenced by the transfer and deposition of atmospheric momentum, transported by upward propagating waves. This momentum perturbs the purely zonal geostrophic flow, which would exist in the absence of any momentum exchange for the case of an atmosphere in radiative equilibrium. In particular, the ageostrophic meridional flow is affected by this momentum exchange, which leads to mesospheric up-welling and down-welling. As a consequence, adiabatic cooling and heating occurs forcing the atmospheric temperature structure away from radiative equilibrium, resulting in a non-radiative equilibrium wind pattern (e.g., Middleton et al., 2001; Becker, 2012). The observed wind, in turn, is a superposition of several atmospheric waves, such as planetary waves (PW), tidal waves, and gravity waves (GW), which are categorized according to their spatial extents and periods.

Large scale PWs are primary formed in the troposphere by topography and diabatic heating. They influence the general circulation by transferring warm air from the tropics to the poles and by returning cold air towards the tropics. Planetary waves with periods of 2, 5, 10, and 16 days and their role in dynamical processes within the MLT and regions above and below have been frequently discussed in literature (e.g., Iimura et al., 2015; Egito et al., 2016; Matthias and Ern, 2018).

Migrating and non-migrating atmospheric tides in the MLT are crucial for understanding the dynamics in the atmosphere, in particular for vertical coupling processes between several atmospheric layers. They serve as a carrier of momentum, which can be deposited in areas far away from their source region (e.g., Pedatella et al., 2012; Yiğit and Medvedev, 2015). Non-migrating tides are generated by longitudinal differences in radial heating (e.g., Hagan and Forbes, 2002) and while propagating

upwards the tidal amplitude grows significantly due to the exponential density decrease. The dissipation of tides contributes to fluctuations in the mean wind flow (e.g., Lieberman and Hays, 1994). For equatorial latitudes, the most dominant tide is the diurnal (24h), but according to the linear tidal theory (Lindzen and Chapman, 1969), at middle and high latitudes, the diurnal tide does not primarily dominate at the MLT. Therefore, at these latitudes it is the semidiurnal (12h) tide which is important, having highest amplitudes during the winter months and during the autumn transition (e.g., Hoffmann et al., 2010; Jacobi, 2012; Pokhotelov et al., 2018).

Primary GWs, which are generated in the troposphere propagate upwards, with the amplitude of the waves increasing exponentially and efficiently transporting momentum and kinetic energy into the middle atmosphere. The main tropospheric source of GWs is the airflow over orographic irregularities, such as mountains, the vertical movements in convection cells, and strong wind shears in combination with jet instabilities. Here, gravity acts as the wave's restoring force against vertical movement. Depending on the propagation direction of the background wind relative to that of the GWs, strong filtering can occur at different heights. For example, during the summer, the mainly eastward directed GWs are able to reach the mesosphere because most of the westward propagating waves get filtered by the westward directed stratospheric background wind. If GWs break at the MLT, they deposit upward transported momentum onto the background wind, which can lead to a wind reversal (e.g., Fritts and Alexander, 2003). The horizontal scale of the associated excitation varies between several tens and several thousand kilometers with associated periods of minutes up to one day (Tsuda, 2014).

Examining the observed wind by decomposing it into its distinct spectral components has been performed by several studies in recent years (e.g, Eckermann et al., 2016; Hysell et al., 2017; Shibuya et al., 2017; Baumgarten et al., 2018). For this study, we use the approach of decomposing the wind according to Stober et al. (2017); Baumgarten and Stober (2019) by applying an adaptive spectral filter technique (ASF). In this technique the decomposition of the observed wind is basically done by adapting the window length for each tidal component and a vertical regularization of the phase slope using the classical harmonic approach:

$$u, v = u_0, v_0 + \sum_{n=1}^{3} a_n sin(2\pi/T_n \cdot t) + b_n cos(2\pi/T_n \cdot t), \tag{1}$$

where $T_n$ takes the values of 24 hours, 12 hours and 8 hours to determine the diurnal, semidiurnal and terdiurnal tide for each wind component. $a_n, b_n$ are the coefficients of the appropriate amplitude. The gravity wave activity is the residuum, which includes all fluctuations different than tides and planetary waves.

GW activity is often expressed in terms of spectra as a function of wave frequencies and wave numbers, which is rather challenging considering the observational limitations. Therefore, Fritts and VanZandt (1993) described an energy spectrum for

the wind velocity, which is composed of a combination of several GWs. Tsuda et al. (2000) defines the total wave energy as the sum of the potential energy and kinetic energy $E_k$ per unit mass, the latter being given by

$$E_k = \frac{1}{2}(u'^2 + v'^2 + w'^2),\tag{2}$$

where $u'$ and $v'$ are the perturbation of the horizontal wind velocity and $w'$ is the vertical wind perturbation to the wave propagation direction. Even with very precise measurements $w'$ is much smaller than the horizontal perturbations and therefore can be and is very often neglected.

To illustrate the different components, Figure 1 shows a decomposition of the observed wind (top) into the mean wind and tidal component (middle) and the GW residual (bottom). The decomposition is shown for the location of Andenes for ten days. Further information and a more detailed description regarding the algorithm can be found in Section 2.

Long-term changes (LTC) in the atmosphere are complex. They are influenced by several factors including fluctuations in solar and geomagnetic activity, which in turn can induce changes in the neutral density together with changes in the zonal directed winds (e.g., Emmert et al., 2008; Stober et al., 2012), or by anthropogenic emissions of greenhouse gases, which affect the troposphere through increased heating and causes cooling in the upper atmosphere (e.g., Beig, 2011; Laštovička et al., 2012). Several studies investigated LTC based on radar measurements for the northern high and mid-latitudes as e.g., Middleton et al. (2001), Portnyagin et al. (2004), Portnyagin et al. (2006), Keuer et al. (2007), Jacobi et al. (2008), Hoffmann et al. (2011), Iimura et al. (2011), and Jacobi et al. (2015). From these studies, meteor radar wind observations show for the last decade seasonal dependent results for the mid-latitudes, with stronger eastward and southward directed tendencies during the autumn and winter, and opposite tendencies during the spring (e.g., Jacobi et al., 2015). For high latitudes, the zonal wind shows a time varying tendency with an overall eastward directed wind during the winter and also an increase of the semidiurnal tidal amplitude. However, large differences are present among these studies which are based on different measurement intervals and different latitudes (e.g., Iimura et al., 2011).

In this study, we present climatologies and the decadal variability of winds, tides, gravity waves, and planetary waves from the northern high latitude location Andenes, and the mid-latitude locations Juliusruh and Tavistock (CMOR). The data are described in Section 2 and the resulting climatologies and decadal climate variabilities for the wind are presented in Section 3 and for diurnal and semidiurnal tides, gravity waves, and planetary waves in Section 4, respectively. The wind and tidal response on an 11 year oscillation is described in Section 5. Section 6 concludes the paper.

## 2 Data

This study uses observations from three meteor radars (MR), which are located at the polar latitude station of Andenes (69.3°N, 16.0°E, Norway), the mid-latitude location Juliusruh (54.6°N, 13.4°E, Germany), and the mid-latitude location of Tavistock, the Canadian Meteor Orbit Radar (CMOR, 43.3°N, 80.8°W, Canada).

The Andenes MR was installed in 2002 and was run with a 15 kW transmitter at 32.55 MHz until May 2008. In May 2008 the system was moved to a new location 4 kilometers away from the original site. Later in 2009, the system was further upgraded to 30 kW transmitting power. In 2011 and 2012 the original antennas were updated and replaced. Since 2012 the system runs in a stable hardware configuration. However, the experiment settings also underwent some changes during this interval. From 2002 to 2015 (October) the radar ran an experiment with a pulse repetition frequency of 2096 Hz and a 3.6 km mono pulse using a 2 km range sampling. In October 2015 the experiment was changed and the system is now operated with a pulse repetition frequency of 625 Hz and transmits a 7-bit Barker code with 1.5 km range sampling.

The time series of the Juliusruh MR is a composite of several different radar systems. From 2002 to 2010 the OSWIN radar was operated in a meteor mode interleaved to its normal MST-radar observations at a transmitting frequency of 53.5 MHz. These measurements were conducted 118 km west of the later Juliusruh MR site. In November 2007 the Juliusruh MR started its operation as dual frequency radar at 32.55 and 53.5 MHz. The experiment settings were similar to the ones in Andenes between 2002 and 2015. From 2014 to 2015 the system underwent several modifications. First, the experiment settings were changed to run the 625 Hz pulse repetition frequency and a 7-bit Barker code with 1.5 km range sampling (Stober and Chau, 2015). From January 2014 until autumn 2014 the transmitter of the Juliusruh 32.55 MHz system was not operating and only the 53.5 MHz system was observing. In spring 2015 the Juliusruh 53.5 MHz radar ceased its operation and the Juliusruh 32.55 MHz system remained operational, but with an increased transmitting power of 30 kW. Since this last modification, the system operates continuously in a stable hardware and experiment configuration.

The CMOR MR provides the longest and most homogeneous MR time series used in this study. The system did run in a more or less unchanged configuration since 2002 as a triple frequency system (17.45 MHz, 29.85 MHz, and 38.15 MHz) near Tavistock, Canada. Observations are carried out with a pulse repetition frequency of 532 Hz using a 11 km mono pulse and 3 km range sampling. The 17 and 38 MHz radars each use a 6 kW transmitter, the 29 MHz system was upgraded from 6 to 12 kW in the frame of the CMOR2 upgrade in May 2009. In this study, we compiled one homogeneous wind data set involving all available data of the triple frequency observations.

In this study, the composites and LTC are based on data sets for the years 2002 - 2018 for each location. The winds are obtained applying a modified version of the all-sky fit (Hocking et al., 2001; Stober et al., 2018) and they have an hourly temporal resolution and partly covering the heights between 70 and 110 km, with a vertical altitude resolution of 2 km. The different atmospheric waves are extracted by an adaptive spectral filter (ASF) (Stober et al., 2017; ?). In this study, we focus on observed mean winds, tides, gravity, and planetary waves. The statistical uncertainties are based on the applied fitting procedure by taking into account full error propagation of the radial wind errors, as well as the number of meteors per altitude and time bin. The resulting uncertainties of the wind vary in the range of 2 - 16 $ms^{-1}$, with larger errors occurring in bins with fewer meteors or at the upper and lower edges of the meteor layer. More information about the experimental setup and the technical specifications for the Andenes and Juliusruh meteor radars, as well as about the wind analysis and the obtained uncertainties for all three radars can be found in Stober et al. (2017, 2018). More technical information about CMOR and CMOR2 are described in e.g., Webster et al. (2004); Jones et al. (2005); Brown et al. (2008).

## 2.1 Homogenization of time series

The instruments used in this study were operational for almost two decades and some meteor radars did undergo substantial maintenance and modifications on the hardware. Most crucial for the wind measurements are the phase calibration and stability, the range sampling and the Doppler measurement. The Andenes and Juliusruh meteor radar were maintained twice a year including a test of the phase match of the cables and antennas. Further, the SKiYCORR software runs a phase test and provides a summary file of the impedance for each channel and day indicating potential problems. In addition, to the regular maintenance the CMOR meteor radar interferometry (phases) is cross validated to optical observations. In particular, meteor showers are monitored with CMOR throughout the year providing another source of information on the phase stability. The Andenes and Juliusruh meteor radar were also checked and cross validated using selected meteor showers during the course of the year.

Both European meteor radars were frequently range and power calibrated using a delay line (Latteck et al., 2008; Stober et al., 2010). The CMOR radar is also routinely checked for potential issues in the range sampling applying various cross calibrations. All systems used the same software package over the complete time span to derive the Doppler velocities to avoid artefacts due to changes in the parameter estimation (e.g., Doppler velocity or the velocity uncertainty).

Before the multi-frequency data sets for CMOR and Juliusruh are compiled, we analyze the winds for each frequency independently and cross-validate the resultant time series. If one instrument shows systematic issues in the wind time series compared to the other instruments and the climatology, this data is flagged and no longer considered in the finally compiled and merged wind time series. The Andenes meteor radar data is campaign wise cross-validated with other meteor radars in Norway.

## 2.2 Adaptive spectral filtering of time series

The ASF provides a wave decomposition of our original observed time series into a daily mean wind, diurnal and semidiurnal tides as well as a gravity wave residuum with an hourly resolution. Here, the gravity wave residuum also includes the terdiurnal tidal component. The hourly resolved time series are then averaged to daily means keeping the error information. The ASF is designed to account for the intermittency of waves, in particular, of tides and mean winds for time periods less than a day. Therefore, we adapt the window length of the harmonic tidal fit to the number of wave cycles. In the first step, we fit the daily mean wind with a window length of 24 hours plus all tidal components. The next step uses the daily mean wind and the diurnal tidal component as boundary to extract the information of the semidiurnal tide and so forth. This procedure is applied as a sliding window along the time series and all wave information (amplitude and phase) for all waves are determined for each time step. The technique is least squares based and, hence, robust against unevenly sampled data or data gaps shorter than the length of the window. Another benefit of the least squares implementation is the error propagation to all derived parameters. Further, we implemented a regularization constraint for the mean winds, diurnal and semidiurnal tide making use of the vertical wavelength information assuming that the mean winds and tidal phase should only show gradual changes within a vertical kernel function of 8 km for the mean winds and 10 km for the tidal phases. The daily mean wind time series (tides and

gravity waves removed) are further analyzed to obtain the planetary wave activity. Therefore, we define a seasonal background wind based on the daily mean time series of $u_0$ and $v_0$ for the zonal and meridional component, respectively;

$$u_0, v_0 = u_m, v_m + \sum_{i=1}^{2} a_n sin(2\pi/T_n \cdot t) + b_n cos(2\pi/T_n \cdot t); \tag{3}$$

Here $u_m$ and $v_m$ are an annual mean zonal and meridional wind, $a_n$ and $b_n$ are coefficients for the seasonal subharmonics with periods $T_n = 365.25/n$ days ($n = 1, 2$). We determine the background wind field for every month at the $15^{th}$ by fitting the above described seasonal model to the daily mean wind time series using a 2-year window centered at the respective month and reconstruct the background wind time series for the other days for each month. The planetary wave activity is then given by subtracting the previously obtained daily mean winds and the reconstructed background wind field. The benefit of this approach compared to other techniques, e.g. smoothing the data or running averages, is that it is more robust against larger data gaps of up to months in length. Another benefit is that due to the long window used for the fitting seasonal peculiarities, e.g, sudden stratospheric warmings, are not affecting the monthly means, but are well captured in the planetary wave activity.

Monthly mean tidal amplitudes, GW and PW activity are derived by computing monthly medians of the available data sets. Thus, the resultant time series contain some data gaps. However, there are still enough data points to estimate a LTC and a potential solar cycle effect for all these waves for each month. The LTC and solar cycle effect are derived by using a linear trend model plus an 11-year oscillation, which is not tied to the F10.7 solar radio flux or the sunspot number. Qian et al. (2019) analyzed WACCM-X and wind observations above Collm (51° N, 13° E) and found that the wind signature is less statistical significant than the temperature response to the solar radio flux. Other studies exploring the stratospheric/tropospheric response to solar forcing indicate a more clear dependence (Rind et al., 2008; Salby and Callaghan, 2006; Lu et al., 2017) on solar activity. At the MLT, the wind seems to be less directly influenced by the F10.7 or sunspot number. Pokhotelov et al. (2018) found almost no correlation between the occurrence of mesospheric echoes at mid-latitudes and the solar radio flux (F10.7), but a clear dependence on the occurrence of these echoes due to meridional winds. Further, Stober et al. (2014) investigated the neutral air density response during the solar cycle 23 and found a phase delay of almost 1 year between the F10.7 proxy and the neutral air density variation. Considering all the aspects above we model the resulting mean winds as:

$$u_{mm}, v_{mm} = a_{u,v} + m_{u,v} \cdot t + a \cdot sin(2\pi/11.0 \cdot t) + b \cdot cos(2\pi/11.0 \cdot t); \tag{4}$$

where $u_{mm}, v_{mm}$ are the monthly mean zonal and meridional components for the mean wind and each wave, $m_{u,v}$ is linear change over the whole period, $a$ and $b$ are the solar cycle components, $a_{u,v}$ is the mean at year 0, and $t$ is the time in years.

## 3 Climatologies and long-term changes of the mean wind

Analyzing long time series always requires an estimate of the associated confidence values of the measured linear changes or other derived parameters. In this study, we conduct a full error propagation to all parameters using the covariance matrices of the fitted functions. Based on this statistical uncertainty we are able to define the 90% and 95% confidence levels given by $x \pm \sigma z$. Here $x$ is a parameter, $\sigma$ is the statistical uncertainty of $x$ and $z$ is a factor, which takes values $z = 1.64$ for the 90% confidence interval and $z = 2$ for the 95% interval, respectively, assuming a Gaussian error distribution. We label the different confidence intervals by dashed (90%) and solid (95%) contours for all derived parameters. We tested our confidence intervals whether they are significant by introducing two null hypothesis. The long term linear changes are tested under the assumption that there is no linear change as null hypothesis, the solar cycle is tested with the null hypothesis that there is no significant solar cycle. However, it is also important to note that there could be a potential autocorrelation in our time series due to the solar cycle. We estimated all confidence levels under the assumptions that the Gauss-Markov theorem holds and our least square estimators are an unbiased solution, viz. that the fit residuals are uncorrelated.

Mean wind climatologies at the MLT are often shown for a particular location or instrument or as averages over different periods. In this study we present climatologies of mean winds, diurnal and semidiurnal tides, and PW, and GW activity covering more than $25°$ latitude from mid-latitudes to polar-latitudes. Thus we are providing a profile of the mean wind systems at the MLT over the northern hemisphere. Furthermore, the data sets span the same observational periods from 2002 to 2018 and the winds are obtained by the same type of analysis.

The mean wind climatologies are shown in Figure 2. Every location shows a distinct seasonal pattern, with eastward directed winds during the winter and a transition/reversal between east- and westward winds during the summer. Meridional winds are northward directed during the winter and southward directed wind during the summer. The zero line transition is shown as a black contour line. The zonal wind pattern indicates two pronounced features when comparing the different latitudes. In winter, the eastward directed winds are much stronger at CMOR with up to $40\ ms^{-1}$ and decrease towards higher latitudes with 6 - $10\ ms^{-1}$. Further, CMOR shows a zero line crossing in the zonal winds around 100 km altitude, which is not seen at Juliusruh and Andenes. During the fall transition, Juliusruh show for a month at altitudes above 95 km westward directed wind. During summer the wind pattern looks rather similar, just the zonal wind reversal altitude increases from the mid-latitudes towards the polar-latitudes by almost 8 - 10 km (June, July, August).

The meridional wind climatology also shows latitudinal differences. During the winter season, the mid-latitudes show northward winds of magnitude $10\ ms^{-1}$. The summertime is characterized by a southward mesospheric jet of 10 - 15 $ms^{-1}$, which is closely related to the zonal wind reversal. The most prominent feature in the meridional winds is the zero line and its altitude variation during the course of the year. At Andenes, northward winds occur only below 90 km altitude and then for only a few months in winter. In contrast, at the mid-latitude stations northward winds are found at all altitudes throughout the winter and southward winds for the summer months. Due to the different lengths of time series compared to other studies these results are only partly consistent with findings of e.g., Yuan et al. (2008b), Kishore Kumar and Hocking (2010), Hoffmann et al. (2011),

Jacobi (2012), Conte et al. (2018), and Lukianova et al. (2018).

Although the climatologies are statistically robust regarding the mean patterns in both wind components, there is a year-to-year variability and also changes over much longer time scales. Figure 3 shows the time-series of the zonal (left) and the meridional (right) winds for the high latitude location Andenes (top) and the mid-latitude locations Juliusruh (middle) and CMOR (bottom). As described in Section 2, especially for Juliusruh, the system modifications resulted in an increase of the altitude coverage due to software and hardware improvements over several years. The seasonal pattern, shown in the climatologies (Figure 2) is even more clearly visible in Figure 4, where the year-to-year variability is more pronounced, by using the seasonal fit removing the PW activity from the time series.

Just by visual inspection of Figure 4 some of the year-to-year variability or LTC becomes visible, e.g. for the years 2003 - 2007 CMOR shows a westward directed wind regime above 100 km during summer, which disappears in more recent years. Furthermore, there is an enhancement of the southward directed winds in Andenes after the year 2015 at altitudes above 95 km.

Monthly changes are estimated using the equation 4 and are shown in Figure 5 for both wind components. The dashed black lines represent the 90% confidence level, and the solid black lines the 95% confidence level. It is rather obvious from Figure 5 that there is no common linear change at all three latitudes, and, thus we discuss each site separately. At Andenes, an enhancement of the westward directed wind occurs during the begin of the year with values of up to 0.3 $ms^{-1}year^{-1}$, as well as for the summer in the area above the transition height. After the fall transition, a small enhancement of eastward winds is found, with values of up to 0.3 $ms^{-1}year^{-1}$ below 100 km. The meridional wind for Andenes shows a pronounced southward directed wind long-term variability, with values of up to 0.5 $ms^{-1}year^{-1}$ above ∼96 km for the winter and above ∼90 km for the summer. The LTC for Juliusruh is less significant, with changes which correspond to an eastward directed tendency during begin of April and May and westward directed below 90 km at June/July. Furthermore, an eastward enhancement below 90 km between September and November, and at the begin of the year above 90 km, with values of up to 0.5 $ms^{-1}$ is found. The meridional component of Juliusruh shows tendencies towards south between January and April and an opposite tendency between May and November. At the location of CMOR, the strongest significant LTC occurs between April and August with an eastward acceleration of the zonal wind, enhancing the zonal jet above 90 km and weakening the westward jet below with values of up to 0.5 $ms^{-1}year^{-1}$. Meridional winds at CMOR show a southward long-term variability between 90 and 100 km at the beginning of the year and some northward accelerations in summer.

The seasonal analysis provides information about the mean zonal and meridional wind for each year and altitude. Figure 6 shows the vertical LTC based on annual mean values. The vertical profiles indicate the linear change per decade of the zonal (red) and meridional (blue) wind. The most significant changes occur at Andenes in both wind components. The mean zonal wind speed is decreasing between 85 - 100 km by up to 3 $ms^{-1}decade^{-1}$. The LTC of the meridional wind reaches values up to 2 $ms^{-1}decade^{-1}$. At mid-latitudes (Juliusruh) the zonal wind shows only a weak change per decade and an eastward acceleration with 0 - 0.5 $ms^{-1}decade^{-1}$. The meridional winds indicate a more pronounced linear tendency. Below 85 km the meridional jet seems to be further westward accelerated, whereas at higher altitudes an eastward acceleration is found.

At CMOR the zonal wind shows almost no long-term variability at all altitudes between 75 - 110 km. The meridional wind indicates a LTC above 90 km altitude corresponding to a northward acceleration of the mean circulation.

## 4 Climatologies and long-term changes of waves

### 4.1 Diurnal tides

The monthly median amplitudes and the associated composites for the tidal 24h-diurnal components are shown in Figure 7 and Figure 8. The seasonal pattern of the diurnal tidal (DT) amplitude shows a rather rapid increase around 100 km altitude and at least during the summer a secondary enhancement around 80 km altitude with values of $\sim$15 $ms^{-1}$. Comparing all three locations, CMOR shows the strongest maximum and strongest mean amplitudes for the zonal diurnal tides, with mean values larger than 25 $ms^{-1}$. This occurs at heights above 90 km and especially between January and April shows a general

enhancement of the zonal diurnal tidal amplitude. Juliusruh reaches maximum mean values of $\sim$25 $ms^{-1}$ only between the late summer and autumn above 100 km. During this time, Andenes also shows the strongest diurnal tidal amplitudes in the zonal direction, but with weaker maximal mean values of up to 20 $ms^{-1}$. The meridional diurnal tidal component at all three locations shows a similar pattern, with enhancements of the amplitudes between summer and winter, for heights above 94 km, where it reaches maximum mean values of over 30 $ms^{-1}$. All locations show a second increase during the summer around 82

15 km, and even higher up for CMOR, with mean values of 15 - 20 $ms^{-1}$. Another very prominent feature of the diurnal tidal amplitudes is related to its polarization relation. At Andenes and Juliusruh the meridional component is significantly enhanced compared to the zonal diurnal tidal amplitude. At CMOR this effect is less pronounced during June - December and reverses in spring, where the zonal diurnal tidal amplitude is much larger compared to the meridional component.

Comparing our climatologies to previous studies reveals some interesting differences. (Portnyagin et al., 2004; Jacobi, 2012)

found a distinct maximum during the summer months and almost no tidal signature in winter at altitudes from 92-98 km. Both studies did also not show the diurnal summer maximum below 82 km. These difference are partly explainable by the different length of the analyzed time series.Portnyagin et al. (2004) could only use a bit more than 1 year of data for the Scandinavian climatology. Jacobi (2012) compiled a climatology from 6 years during solar min conditions. However, the ASF decomposition of tides and mean winds considering the intermittent behaviour of the diurnal amplitude and phase may also play a role.

The diurnal tidal phases are shown in Figure 9. The phases are referenced to a longitude of $13°$ east. The white contour line labels phase jumps and zero phases. The CMOR phases are shifted as if they would have been observed at the CMOR latitude, but at the above mentioned longitude in the European sector. The diurnal tidal phases show a distinct seasonal pattern and latitudinal differences. Throughout the year there are substantial changes in the phases at a given altitude, in particular at the polar latitudes during the winter months, the phases undergo phase drifts of several hours within a month.

Based on the long-term series, Figure 10 indicates the interannual LTC for the diurnal components. For the locations of Andenes and Juliusruh, the diurnal component shows small, but significant tendencies. During the summer at Andenes, a westward directed amplitude gradient is present in the westward wind regime below 85 km, with values of up to 0.3 $ms^{-1}year^{-1}$. Furthermore, there is a northward directed wind amplitude change during the fall at around 100 km. At the location of Julius-

ruh, changes take place in the zonal component during the winter with a tendency towards a decreasing diurnal tidal activity above 90 km. However, at Andenes and Juliusruh the zonal and meridional diurnal tidal amplitudes show only rather small changes from 2002 to 2018. At CMOR changes emerge between 82 and 100 km in January. During the early winter, the LTC shows an increasing diurnal tidal amplitude activity with values up to 0.4 $ms^{-1}year^{-1}$ for the zonal component and almost no

change for the meridional component. During the summer months, the LTC points towards a decreasing tidal amplitude with up to 1 $ms^{-1}year^{-1}$ for heights above 100 km. Meridional tidal diurnal amplitudes at CMOR exhibit only small changes.

## 4.2  Semidiurnal tides

The 12h-semidiurnal tide is the most dominant wave in the MLT throughout the year at mid- and high latitudes. The time series of semidiurnal tidal (SDT) amplitudes is presented in Figures 11 and the SDT climatology is given in Figure 12. SDT ampli-

tudes are usually larger compared to DT amplitudes, and reach at the mid-latitudes for the zonal wind component maximum mean values of ∼30 - 40 $ms^{-1}$, and for the meridional component maximum mean values of 20 - 40 $ms^{-1}$. In general, the semidiurnal tidal components at all locations show similar seasonal pattern. SDT amplitudes increase with increasing heights and reach maximum values around 100 km altitude. The seasonal pattern of the SDT shows a very similar morphology through-out the year at all three MR sites. There is a winter maximum, a spring minimum and a second amplification during September

- October and a second minimum in November. At Andenes the SDT amplitude reaches mean values for both components of up to 30 $ms^{-1}$. The highest SDT amplitudes are seen at the mid-latitude station Juliusruh during the winter months with values of up to 40 $ms^{-1}$. In contrast, the fall transition reaches its highest SDT amplitudes of ∼40 $ms^{-1}$ (zonal component) at CMOR.

At Andenes and Juliusruh the zonal and meridional wind components indicate similar values for amplitudes and occurrence

of the SDT. Comparable amplitudes are present during the winter months at CMOR. However, the fall transition above looks slightly different for the CMOR MR. The zonal SDT amplitude appears to be larger than the meridional component.

Figure 13 shows the phase behaviour for the SDT. The seasonal pattern indicates an asymmetry and rapid phase change during the fall transition and the winter months. During the spring transition the phases also are altered, but less prominent compared to the fall and winter time. The phases also reflect the seasonal asymmetry similar than for the amplitudes of the

SDT. Further, it appears that latitudinal differences between Andenes and Juliusruh are small, whereas the phase differences to the CMOR latitude are much more significant. SDT phases also show continuous phase changes throughout the year. During the fall transition the phase changes within a month by more than 6 hours at all three latitudes. However, also the winter time is characterized by drifting SDT phases within a month.

The LTC for the semidiurnal tides is shown in Figure 14. At Andenes a significant change emerge above 90 km during the

winter (November, December) showing a rather strong decrease of the SDT with amplitudes of 1 $ms^{-1}year^{-1}$. Additionally, a significant enhancement of the SDT occurs during the autumn transition showing an increase of up to 1 $ms^{-1}year^{-1}$. Similar pattern for the summer also occur for Juliusruh. This behavior is not reflected at CMOR. There, it appears that the SDT amplitudes in November are further increasing in the zonal and meridional component. CMOR also exhibits a significant increase in the wintertime (December-February) of SDT amplitudes above 90 km.

## 4.3 Planetary and Gravity waves

The planetary wave activity is estimated as residual between the daily mean winds, as obtained from the adaptive spectral filtering, and the seasonal fit shown in eq. (3). The seasonal fit provides a robust estimate of a background wind field for every day of the year and each wind component. The zonal and meridional wind residuals can be written as $u'$ and $v'$ and are considered as a good proxy of a planetary wave amplitude. However, this method does not allow to distinguish between a planetary wave like oscillation and a SSW event, which typically lasts 3-5 days at the MLT. However, Matsuno (1971) already pointed out that PWs play a major role in the evolution of SSWs. Figure 15 shows the PW energy. All three locations show striking enhancements during the winter, especially during years when a major sudden stratospheric warming (red arrow) takes place. During the years with a major sudden stratospheric warming event, the PW energy appears to be increased and takes values of up to 300 $m^2 s^{-2}$ in the winter months. Minor sudden stratospheric warmings (green arrow) show also an increase of the PW energy, but usually weaker than during years with a major sudden stratospheric warming. Even for the year 2016, where we found an exceptional circulation pattern at the MLT an enhancement of the PW energy is present Stober et al. (2017), although this year did not show the evolution of a typical SSW (Matthias and Ern, 2018). For the rest of the year, the PW activity is comparatively low, with sparse enhancements observed at CMOR.

In Figures 16 and 17 the long-term observations of kinetic gravity wave energy (GW) and the corresponding GW energy climatology are presented. The general seasonal pattern for all three locations appears to be quite similar. An enhancement of the kinetic GW energy with increasing heights is noticeable, as well as a seasonal pattern with increased GW energies between the autumn transition and the end of the winter, with values of up to 400 $m^2 s^{-2}$. Below an height of $\sim$82 km during the summer there is a secondary enhancement, which is especially noticeable at Andenes and Juliusruh. At that time, values of up to to $\sim$150 $m^2 s^{-2}$ are recorded for Andenes, and up to $\sim$250 $m^2 s^{-2}$ for Juliusruh.

## 5 Wind dependencies on an 11-year oscillation

For long-term wind data which exceeds the period of a solar cycle it is advantageous to consider the influence of an 11-year oscillation on the wind. Figure 18 visualizes the impact of an 11-year oscillation on a seasonal basis. All three stations show nearly no changes in the meridional component, while the zonal winds appear to be highly responsive to the solar activity during the summer around 80 km and during the winter. At the equinoxes, the zonal wind component is unaffected by the 11-year modulation. During the summer, all three locations show an 11-year oscillation with an amplitudes response between 3 - 5 $ms^{-1}$ below 82 km.

In addition to the annual profile, Figures 19 and 20 show seasonal linear influences of solar radiation on the tidal components. The influence of the 11-year oscillation on the diurnal tides is shown in Figure 19. Andenes and Juliusruh show no changes in the zonal component, while the 11-year oscillation in CMOR becomes prominent above 90 km. For the meridional component, only Andenes and CMOR are affected above 94 km during the summer months, by values of up to 4 $ms^{-1}$. For the semidiurnal tides (Figure 20) all locations show for both components enhancements during and after the autumn transition above $\sim$90 km, which last at CMOR until the spring. These enhanced values are remarkable because during the time after the autumn transition

the tidal amplitudes are quite low (see Figure 12), which indicates an increased response to the solar cycle for this period of the year. The modulation of the semidiurnal tidal amplitudes due to the solar cycle forcing ranges between 3 and 6 $ms^{-1}$ for the winter season. Considering that some of the previous tidal climatologies are compiled during different phases of a solar cycle explains some of the discrepancies (Jacobi, 2012; Pokhotelov et al., 2018).

The phase information for the solar cycle can be found in the appendix. The phase is referenced to the year 2002. The yellowish to light orange colour indicate a zero phase shift compared to the reference year and corresponds to the maximum solar activity during solar cycle 23 (e.g., F10.7 or sunspot number). Phases that are outside the 90% confidence interval are shaded and should not be treated as reliable due to the weak signal. The phase behaviour itself seems to be rather complex and depends on season and altitude. The phase pattern of mean winds show also a strong latitude dependence in both wind

components. Only the winter season exhibits similarities in the response to the solar cycle of the sun with respect to the phase behaviour. There is a certain coherence of the phases between the latitudes in particular for the semidiurnal tide, which indicates a pronounced phase offset between September/October and December/January, which requires further investigation. However, the diurnal tide is basically only affected at the CMOR station and shows phases close to zero corresponding to a more or less direct response to the solar forcing. A more detailed discussion of the phase behaviour and the potential causes requires

modelling and is beyond the scope of this paper.

## 6   Discussion

We have used meteor radar observations to characterize the mesospheric and lower thermospheric (MLT) winds, tides, gravity waves, and planetary waves for the northern high latitude site of Andenes and the northern mid-latitude sites of Juliusruh and CMOR. Based on measurements between the years 2002 and 2018, long-term changes (LTC) were estimated for winds and

tides at each location. Depending on the length of the data series, the latitudinal location and the observed heights, long-term tendencies can differ significantly with latitude.

For the mean zonal and meridional wind, the typical wind pattern occurs with eastward directed winds during the winter and a switch from westward to eastward winds during the summer. The transition heights were located at lower heights for

the mid-latitude locations. Changes between northward directed winds in the winter and southward winds during the summer were apparent from all the measurements. Furthermore, above 100 km occurs only for CMOR after the autumn transition a westward directed wind field, which lasts until the spring transition. These climatologies fit generally to model studies made by e.g., Jacobi et al. (2009) and Geißler and Jacobi (2017), or to the results of remote sensing instruments by Schminder and Kürschner (1994). However, some of these studies show smaller differences in the wind values than we find, which we ascribe

to different time series or disparities in the window fit length.

Based on annual mean values, the winds in the MLT over Andenes show a tendency of decreasing amplitude for the zonal and the meridional component. In contrast, the mid-latitude locations show weaker tendencies or only increasing tendencies

above a certain altitude. Stronger differences occur when comparing seasonal tendencies for each location, where in some cases opposite tendencies for the same height and same season can occur. Comparing these tendencies with previous studies, differences are to be expected based on differently used time-series and on different averaging periods. Enhancements or weakenings of the mean zonal wind is also expected to take place due to several geophysical processes, such as the quasi-biennial-oscillation or the El-Niño -Southern Oscillation, which are not incorporated in some studies.

In Hoffmann et al. (2011) long-term tendencies were measured based on medium frequency meteor radar for the location of Juliusruh. They found a similar increasing tendencies during the autumn, but a different tendencies during the spring. This difference may due to the particular time series they used, namely between 1990 until 2010. In the work by Jacobi et al. (2015), LTC were estimated, for the mid-latitude meteor radar station Collm (Germany) for the years 2004 until 2014. They used monthly mean meteor measurements and found tendencies similar to our work for the winter through to the summer months for both wind components. However, they reported an opposite LTC for the meridional component during autumn compared to our results. Using the model MUAM, (Geißler and Jacobi, 2017), also shows the northward tendency during the summer for both mid-latitude MRs, based on trends over a 37 year period. In addition, they found a strong opposite LTC for summer at Andenes.

Concerning tides, we find that the observed SDT component dominates over the DT component at Andenes and Juliusruh but reaches nearly similar zonal amplitudes for the lower latitude location of CMOR. The amplitudes of the meridional diurnal component exceeds the value of the SDT for heights above 100 km. The diurnal component is characterized by a second enhancement during the summer, while the SDT component shows an increase in amplitude during the autumn transition at all locations.

The amplitudes and the seasonal occurrence of tides, especially the SDT, corresponds well to an earlier study made by Manson et al. (2009). Their work covered one year with the SDT and DT reported for several northern latitude locations. Similar to the case for the winds, the seasonal LTC pattern differs by location. While for the tidal components Andenes and Juliusruh show similar changes, CMOR shows somewhat opposite tendencies. Similar climatologies for the SDT tides were found at the latitude of ∼40°N based on model results and lidar measurements in several earlier studies (e.g., Yuan et al., 2008a). Later, Pokhotelov et al. (2018) showed agreement between model data and radar SDT tidal measurements for the locations of Andenes and Juliusruh. For diurnal tides, Portnyagin et al. (2004) found similar amplitudes and also a small enhancement during the summer at around 80 km based on medium frequency radar measurements of the diurnal tides between 1990 and 2000.

The climatology of tidal phases for the DT and SDT point out that the tidal phases are not very stable at the MLT and more or less continuously changing throughout the course of the year. In particular, the rapid phase changes during the fall transition and the winter months (DJF) for the SDT are critical for many other analysis using long windows to determine tidal features. Typically, such long windows are used to separate the lunar tide from the SDT (Chau et al., 2015; Conte et al., 2017). However,

already Fuller-Rowell et al. (2016) pointed out that the phase stability is highly important in such an analysis. They found a lunar tide in model data (Whole Atmosphere Model - WAM) as a result of a drifting phase of the SW2 and TW3 tide during an SSW.

For each of the three locations in our study, the planetary wave energy shows abnormally high peak values during the winter when sudden stratospheric warming also is present. According to Matsuno (1971) these warmings are caused by the interaction of upward propagating planetary waves. The values we find for the planetary wave energy correspond well to earlier studies (e.g., Tsuda et al., 1988) with similar values for the kinetic energy reported by Dowdy et al. (2007). The kinetic gravity wave energy for each location shows larger values at higher altitudes and also during the winter, with values of up to 400 $m^2/s^2$. The summer gravity wave energy enhancement, which occurred in Juliusruh at around 80 km can also partly seen with the use of medium frequency radar data. It is even more apparent with the use of model data (Hoffmann et al., 2010).

The 11-year oscillation is found to affect both the observed winds and tides. The strongest influence is on the zonal wind during the solstices. A study made by Keuer et al. (2007) suggests that for the location of Juliusruh, the strongest influences of solar radiation on the zonal wind should be at 80 km than above during the winter, as well as, nearly similar influences for all heights during the summer. Their work suggests that the meridional component should show no impact from solar radiation on the winds. Both findings correspond well to our results. For the tidal diurnal component, particularly at the lower mid-latitude location of CMOR, there is a strong influence from the 11-year oscillation for heights above $\sim 95$ km, while for the SDT components all three MR show a noticeable response to the 11-year oscillation during the winter for heights above 90 km.

## 7  Conclusions

Measuring long term climatologies (LTCs) in the atmosphere requires continuous and consistent observations. In this study, we analyzed observations from three MRs at Andenes, Juliusruh, and CMOR (Canada) at mid- and high-latitudes to obtain LTC in mean winds, diurnal and semidiurnal tides, gravity waves and planetary waves and their latitudinal dependence for the time period between January 2002 and December 2018.

The focus of this study is to characterize the LTC and solar cycle effects on mean winds, atmospheric tides, gravity wave and planetary wave energy at three different latitudes. Our results demonstrate that it is valuable to sustain continuous observations at the MLT region at several locations as there is no common LTC or solar cycle response. Although we provide confidence levels with our measurements, the uncertainties depend on the chosen time windows. However, the very long data sets used in our study shows that there is a significant year-to-year variability.

Our main specific conclusions are:

- Mean wind climatologies show similar patterns between the mid- and high latitudes. However, there is a clear latitudinal dependence of the summer zonal mesospheric jet reversal altitudes from westward to eastward winds, which increases

with increasing latitude. There are also remarkable differences in the eastward zonal winds during the winter time (December - February), which decreases with latitude as well. However, only the Canadian MR shows a zonal wind reversal to westward winds above 100 km altitude. Meridional wind climatologies also reflect the latitudinal dependence with northward winds during winter and southward winds in summer. In particular, the magnitude of the southward wind increases with decreasing latitude and the altitude of the meridional jet corresponds to the altitude behavior of the summer zonal wind reversal.

– The linear change of the zonal and meridional seasonal winds indicate different latitudinal tendencies for each month and component. The most prominent changes are the southward acceleration of the meridional winds at Andenes, the northward acceleration and, thus, weakening of the southward meridional winds at Juliusruh from June to September. CMOR shows the strongest linear response in the zonal wind component with an intensifying summer eastward jet above 84 km and a weakening of the zonal westward winds below.

– The yearly mean winds show only weak linear changes at CMOR and Juliusruh. At Andenes, the yearly mean wind speed seems to become more southward and westward with altitude.

– Diurnal tides show a strong polarization between the zonal and meridional component. Above Andenes and Juliusruh the meridional tide amplitude exceeds the zonal component. The diurnal tide shows only a weak latitudinal dependence of the meridional component, but a significant increase of the zonal amplitude at the latitude of CMOR. Diurnal tides indicate almost no significant linear changes at the investigated latitudes.

– The climatology of the semidiurnal tide shows the highest amplitudes at mid-latitudes above Juliusruh and a similar pattern at all latitudes. The semidiurnal tide shows a similar pattern, regarding occurrence and magnitude, of the zonal and the meridional component. Only during the fall transition above the CMOR MR does the semidiurnal tide not show comparable values in amplitude and occurrence time. During September the zonal amplitude exceeds the meridional component.

– Semidiurnal tides show latitudinal dependent linear responses. Above Andenes and during the winter months (November, December) the SDT amplitude decrease with about $10\ ms^{-1}decade^{-1}$ amplitude above 90 km altitude. The mid-latitude station Juliusruh exhibits almost no significant linear change of the SDT. The mid-latitude station CMOR shows the most significant linear changes of the SDT. During the winter months (November, December, January) SDT amplitudes increase by $5\ ms^{-1}decade^{-1}$. Further, SDT amplitudes during the fall transition (October) seem to be further weakening.

– The climatology of the tidal phases for the diurnal tide and the semidiurnal tide are not very stable. They change continuously through the year. Especially, during the fall transition and the winter occur phase changes for the semidiurnal tide.

– The planetary wave activity shows a large year-to-year variability and latitudinal dependence with the strongest activity at the polar latitudes. Juliusruh and CMOR MR indicate a weaker mean activity compared to Andenes.

– The gravity wave activity also shows a distinct seasonal pattern at all three latitudes with a maximum during the winter months (December, January, February) and late summer (September) above 90 km. Andenes and Juliusruh exhibit a secondary much weaker enhancement in June, July, August below 80 km altitude. CMOR shows a significant increase in the GW energies at higher altitudes compared to the other two stations.

– The mean winds also exhibit a significant amplitude response to an 11-year oscillation. In particular, the zonal mean winds show a characteristic seasonal solar cycle effect. During summer all three stations exhibit an 11-year oscillation with an amplitude of 3 - 5 $ms^{-1}$ in the zonal component below 82 km altitude. The winter months (November, December, January, February) show a solar cycle response below 82 km at mid- and high latitudes and from November to December a relevant solar cycle amplitude between 84 - 95 km at Andenes and CMOR.

– The solar cycle response to the DT is less prominent. Andenes shows some weak amplitude modulation in the meridional component above 90 km between April and November. Almost no solar cycle effect is visible above Juliusruh. CMOR shows the strongest solar cycle effect in both wind components during summer above 95 km altitude and in the zonal component from January to April.

– The SDT exhibits a clear 11-year response at mid- and high latitudes. The SDT the zonal and meridional winds show similar pattern of the confidence levels and amplitudes. All three stations exhibit a strong solar cycle amplitude of 5 - 8 $ms^{-1}$ from October to November and in the altitude range between 84 - 100 km. The Canadian station presents also a significant change from January to March above 100 km.

– DT and SDT phases show a characteristic seasonal behaviour. The temporal evolution of the phases indicates continuous changes throughout the course of the year. SDT phases show rapid phase changes during the fall transition and at polar latitudes during the winter months (DJF). The mean phase behaviour as well as the continuous changes should be considered analyzing the lunar tides.

– Mean winds, DT and SDT show a seasonal dependent solar cycle effect and considerable different seasonal phase responses to the solar forcing. In particular, the SDT fall transition is characterized by an anticorrelation in September/October to the solar activity, whereas the winter months (DJF) seem to respond more directly to the solar forcing (e.g., F10.7 or sunspot number).

*Data availability.* The Andenes and the Juliusruh radar data are available upon request from Gunter Stober (stober@iap-kborn.de).

The CMOR radar data are available upon request from Peter Brown (pbrown@uwo.ca).

*Competing interests.* The authors declare that they have no conflict of interest.

*Authors contributions.*

SW wrote the manuscript with input from all authors. Furthermore, all co-authors contributed to the data interpretation. GS

provided the high-resolution meteor wind data analysis for all stations and ensured the operation of the Andenes and Juliusruh meteor radar. PB ensures the operation of the CMOR meteor radar.

*Acknowledgements.*   This work was partly supported by the WATILA Project (SAW-2015-IAP-1 383) and partly by the Deutsche Forschungsgemeinschaft (DFG, German Research Foundation; project no. LU1174, PACOG as part of the MS-GWaves research unit). Furthermore, we
5   acknowledge the IAP technicians for the technical support. We are thankful for discussions with Peter Hoffmann.

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

## Appendix A:  Appendix

Beside the amplitude information of the solar cycle fitting (Figure 18 - 20), we also computed the phase information. These

are shown for the wind component in Figure A1, for the diurnal tidal phase in Figure A2 and the semidiurnal tidal phase component in Figure A3. The shaded areas are not significant. It figured out to be a very complex situation of all the different features as described in Section 5. Some of them seem to be correlated with the solar cycle and other anticorrelated. The phase behavior reflects a very complex time and altitude pattern that also be observed in the solar cycle amplitude plots.

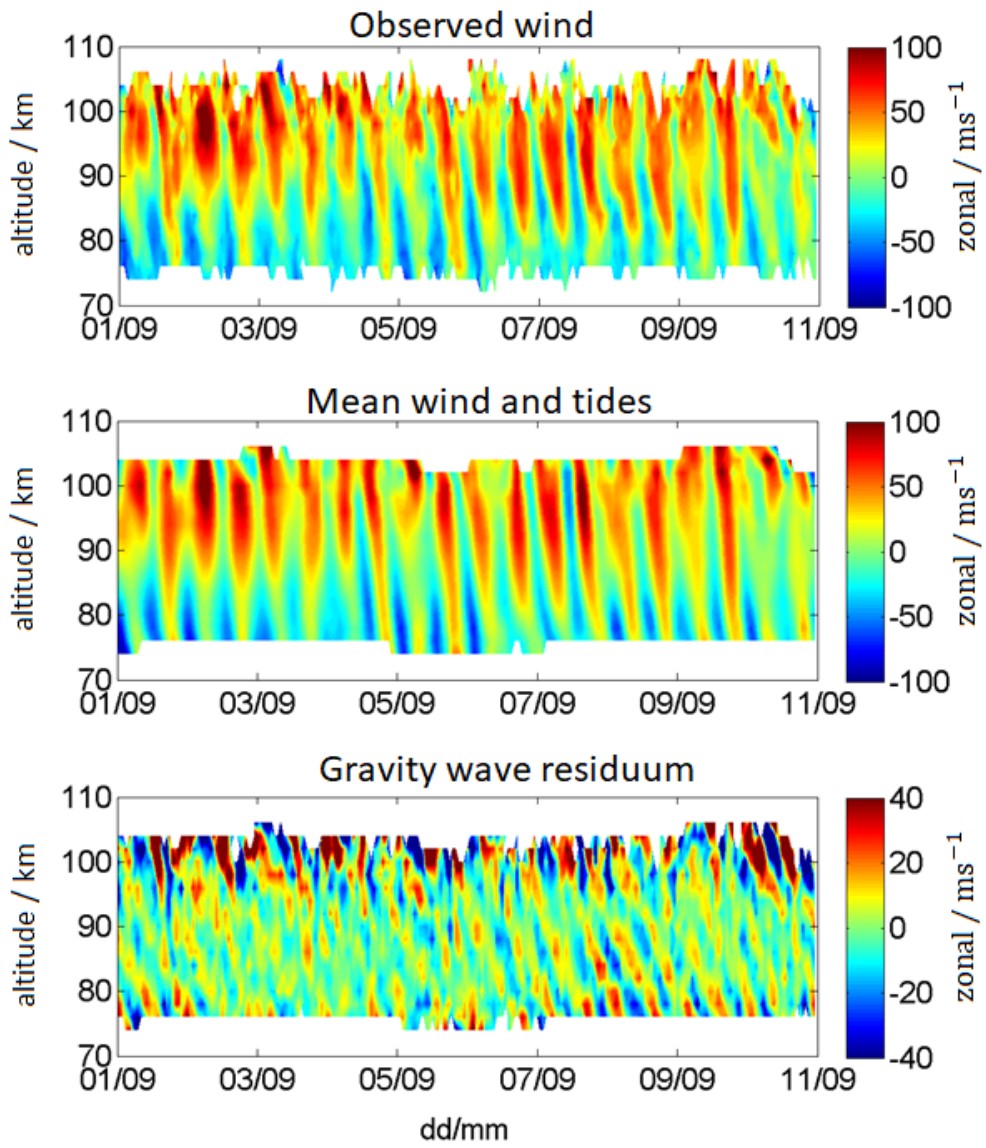

**Figure 1.** Decomposition of the observed wind (top) into the mean wind and tidal component (middle), and the gravity wave residuum (bottom) for Andenes 01/09/2017 - 11/09/2017. Note the different labels of the colorbar.

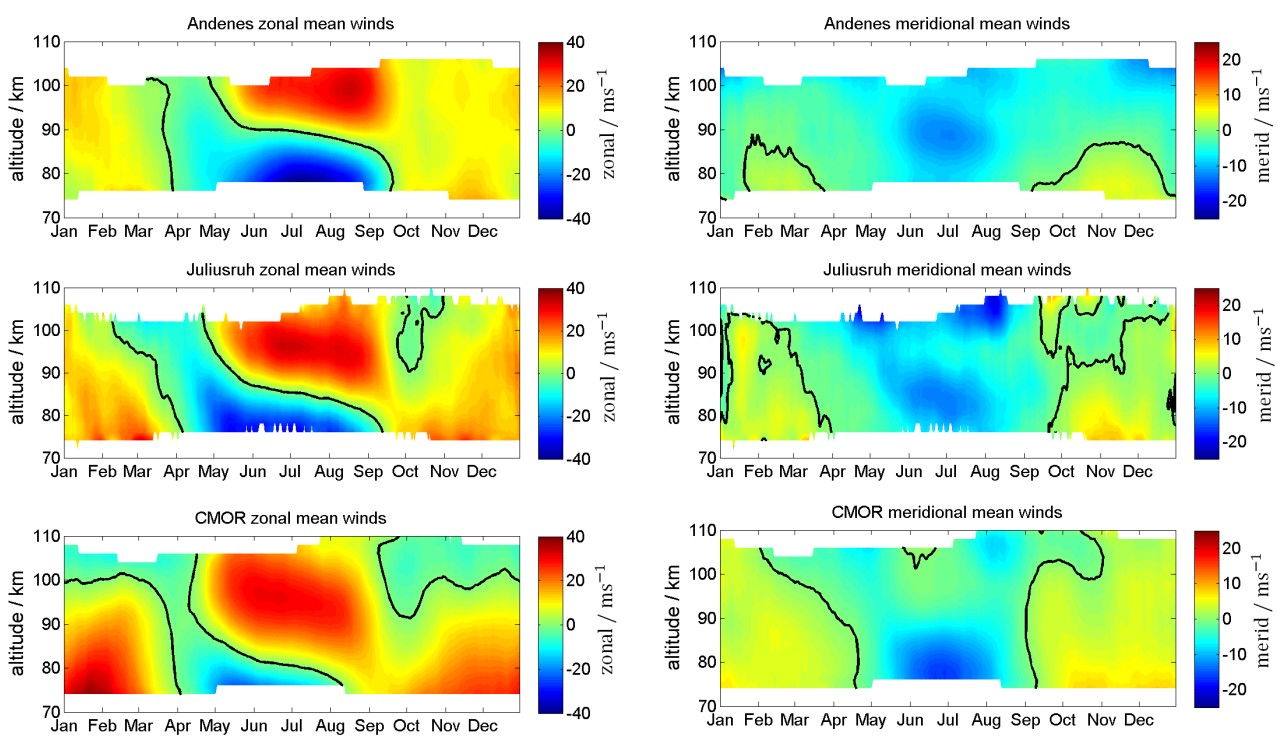

**Figure 2.** Composite of zonal (left) and meridional (right) wind component for the Andenes (top), Juliusruh (middle), and the CMOR (bottom). The black line corresponds to the wind reversal. Note the different labels of the colorbar.

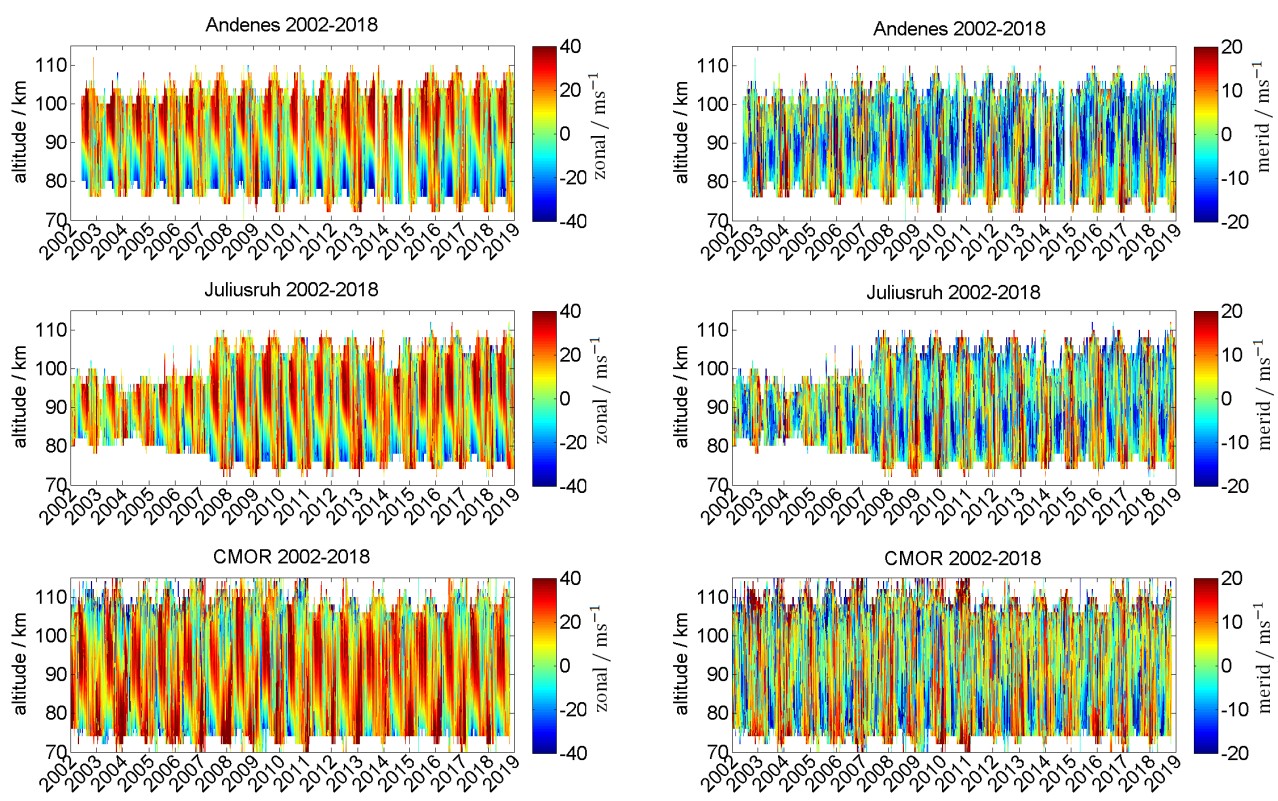

**Figure 3.** Observed zonal (left) and meridional (right) wind components for Andenes (top), Juliusruh (middle), and CMOR (bottom) for the according the location available data series. Note the different labels of the colorbar.

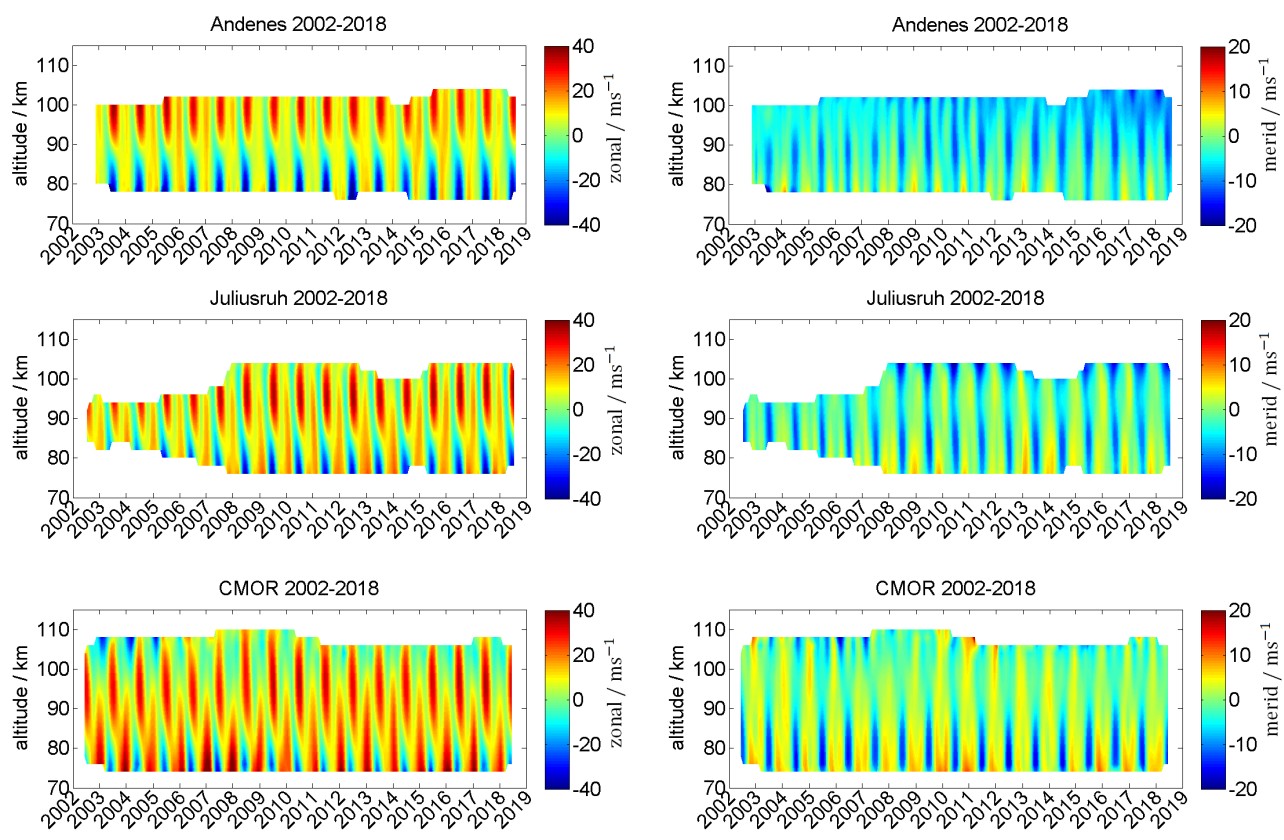

**Figure 4.** Seasonal mean zonal (left) and meridional (right) wind components for Andenes (top), Juliusruh (middle), and CMOR (bottom) for the according the location available data series. Note the different labels of the colorbar.

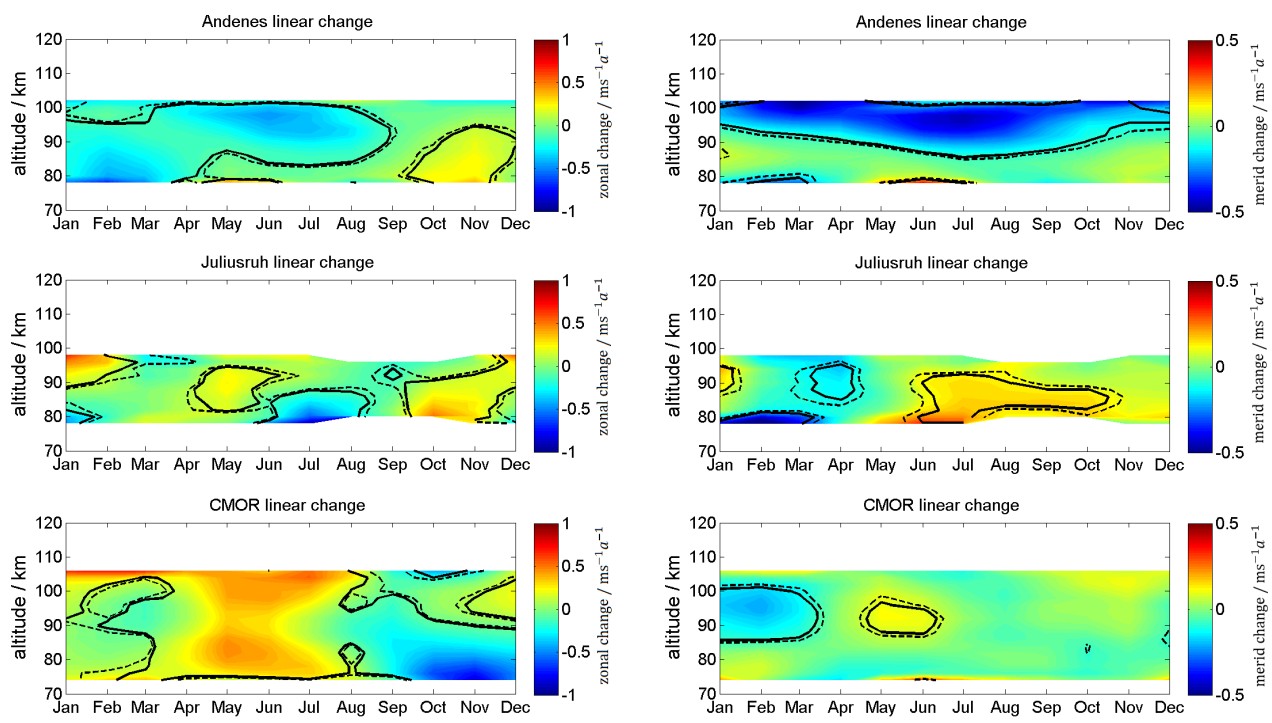

**Figure 5.** Linear long-term changes of zonal (left) and meridional (right) wind for Andenes (top), Juliusruh (middle), and CMOR (bottom). Note the different labels of the colorbar. The solid black lines corresponds to 95% significance, the dashed black lines to the 90% significance.

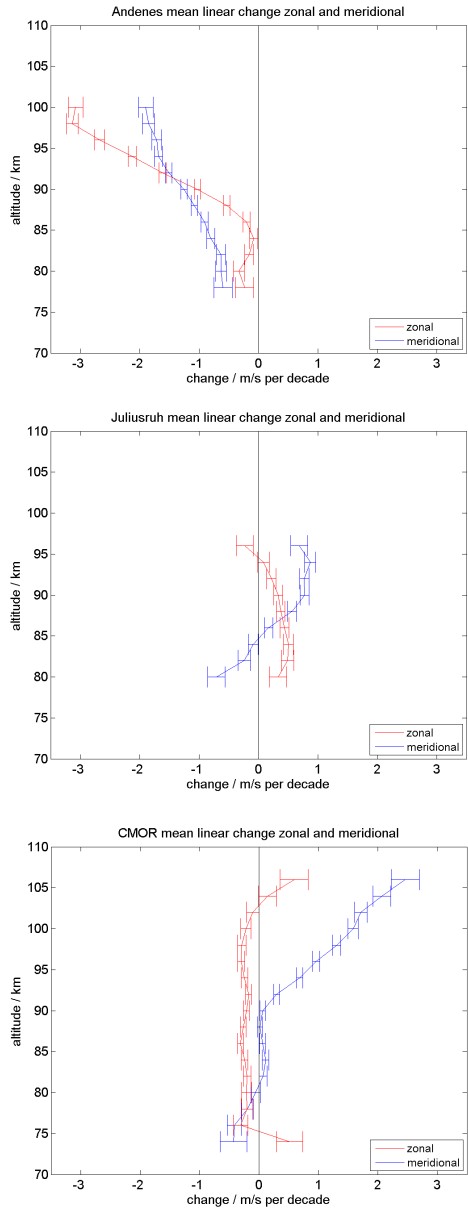

**Figure 6.** Linear long-term changes of zonal (red) and meridional (blue) wind, based on annual values for Andenes (top), Juliusruh (middle), and CMOR (bottom). The errorbars corresponds to the statistical variance.

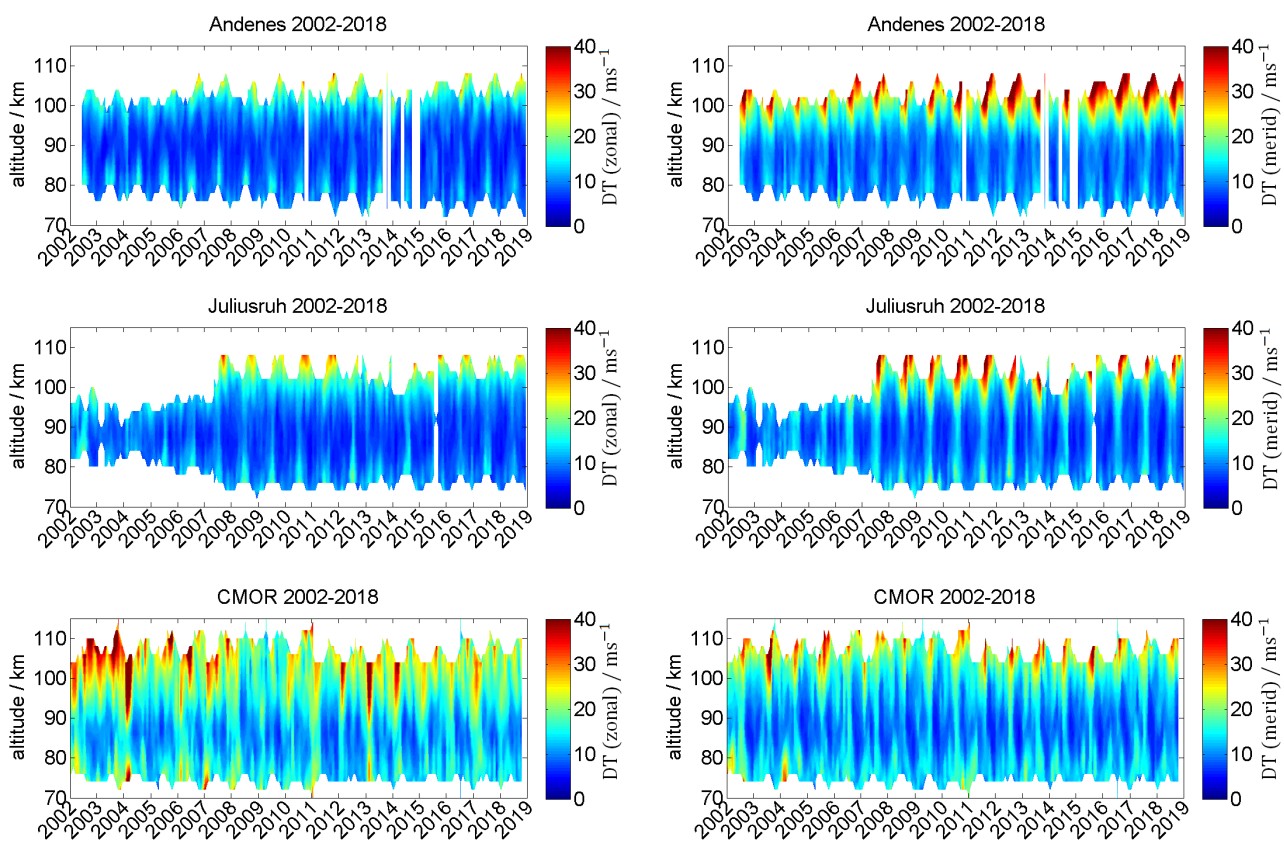

**Figure 7.** Time-series of the zonal (left) and meridional (right) diurnal tidal component for Andenes (top), Juliusruh (middle), and CMOR (bottom).

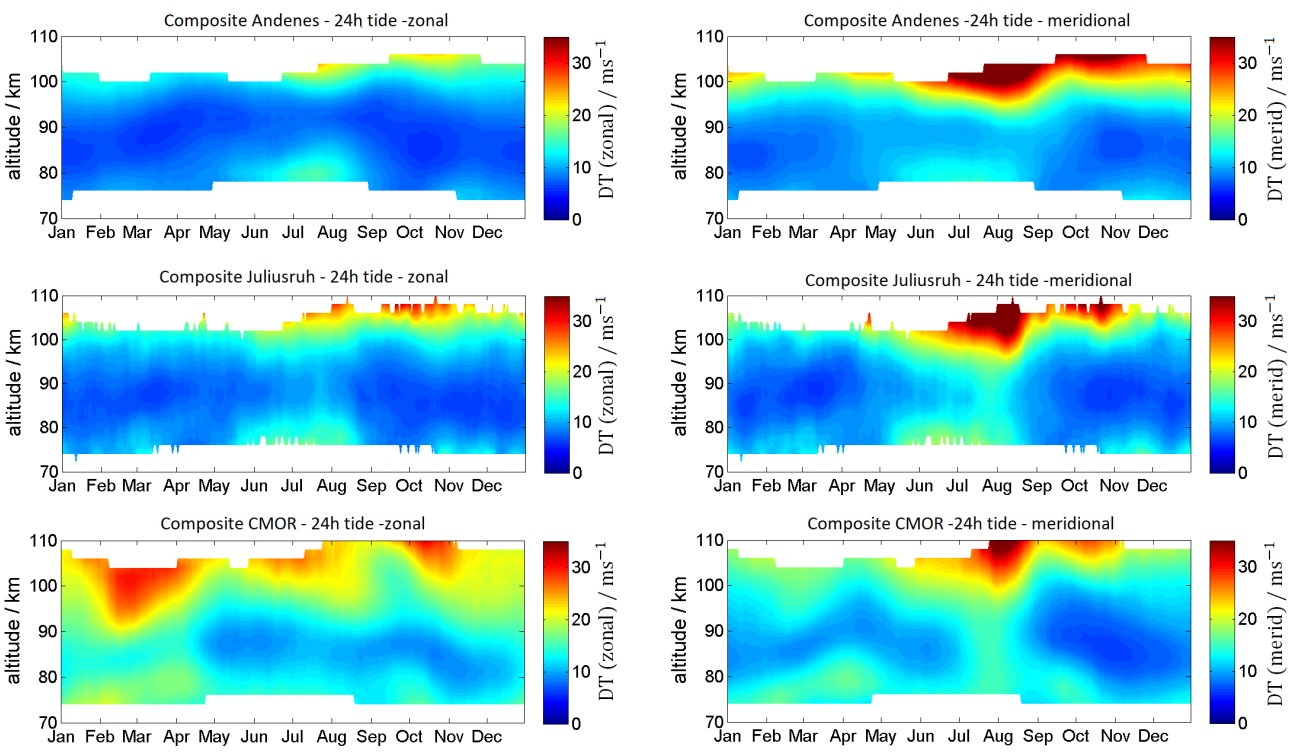

**Figure 8.** Composites of the zonal (left) and meridional (right) diurnal tidal component for Andenes (top), Juliusruh (middle), and CMOR (bottom).

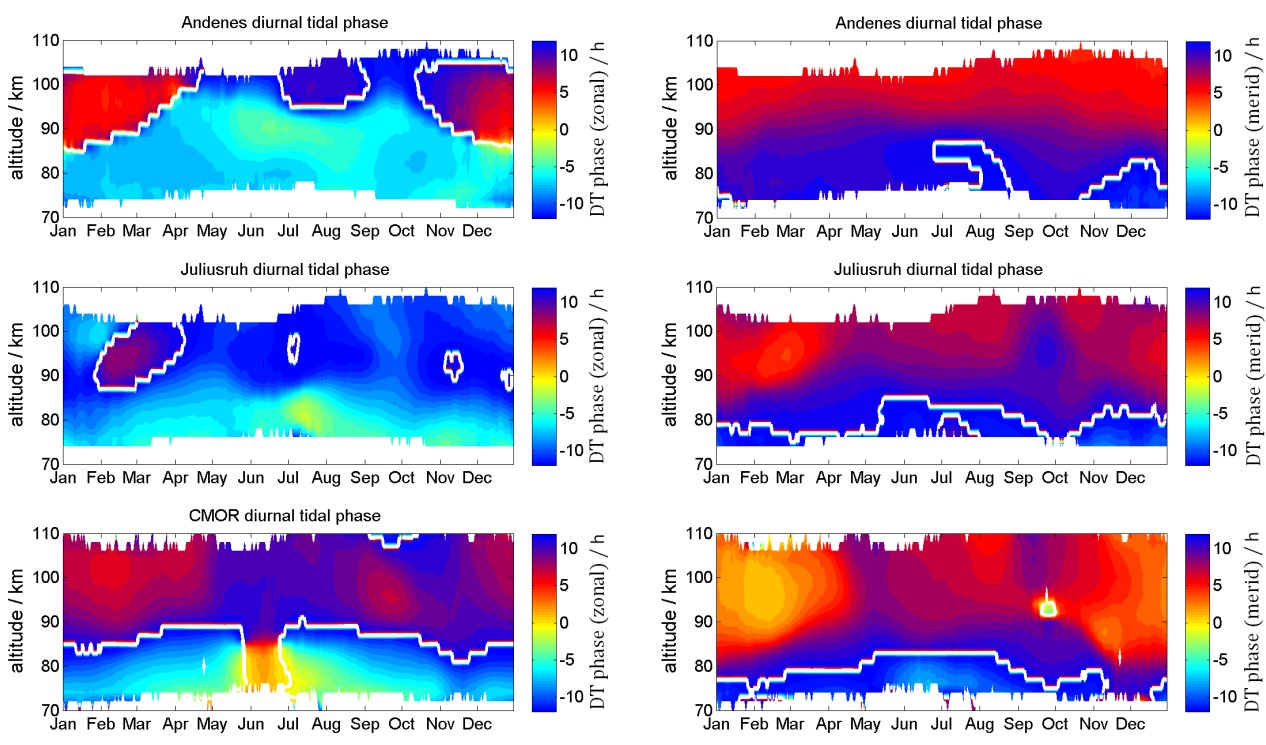

**Figure 9.** Composites of the zonal (left) and meridional (right) diurnal phase information for Andenes (top), Juliusruh (middle), and CMOR (bottom).

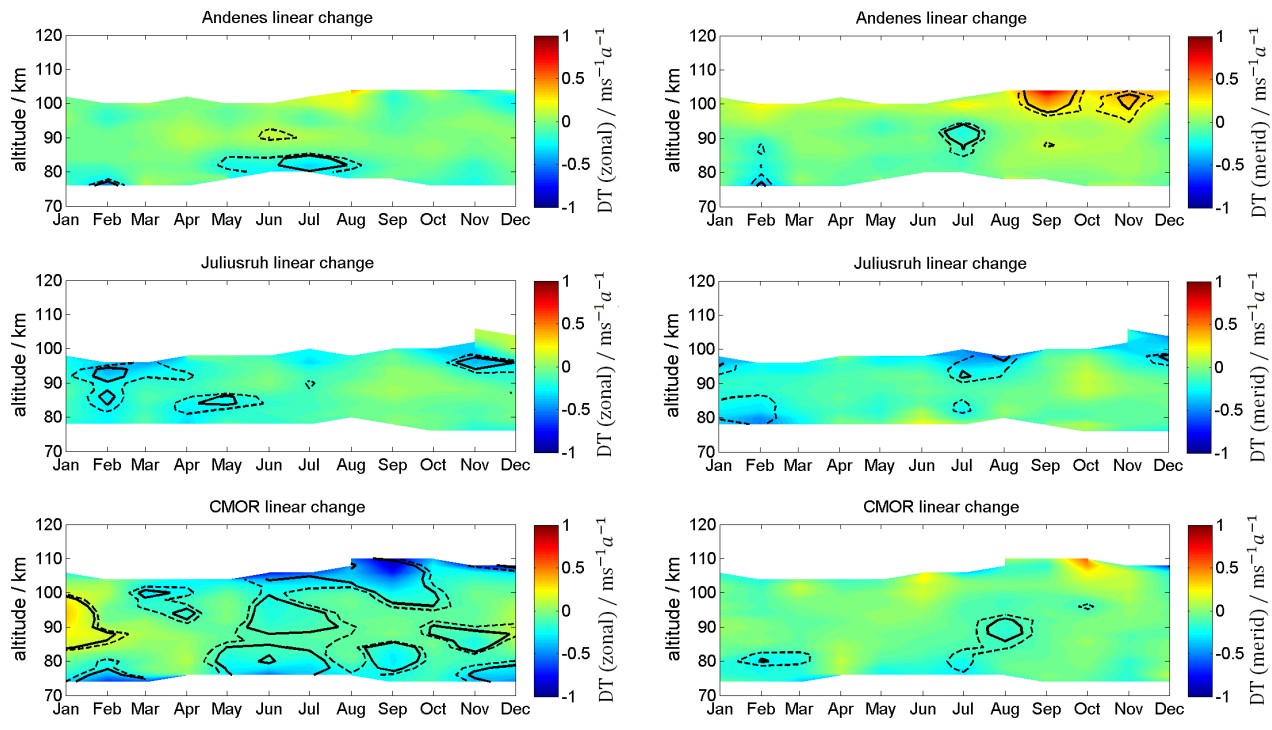

**Figure 10.** Linear long-term changes of zonal (left) and meridional (right) diurnal tidal component for Andenes (top), Juliusruh (middle), and CMOR (bottom). The solid black lines corresponds to 95% significance, the dashed black lines to the 90% significance.

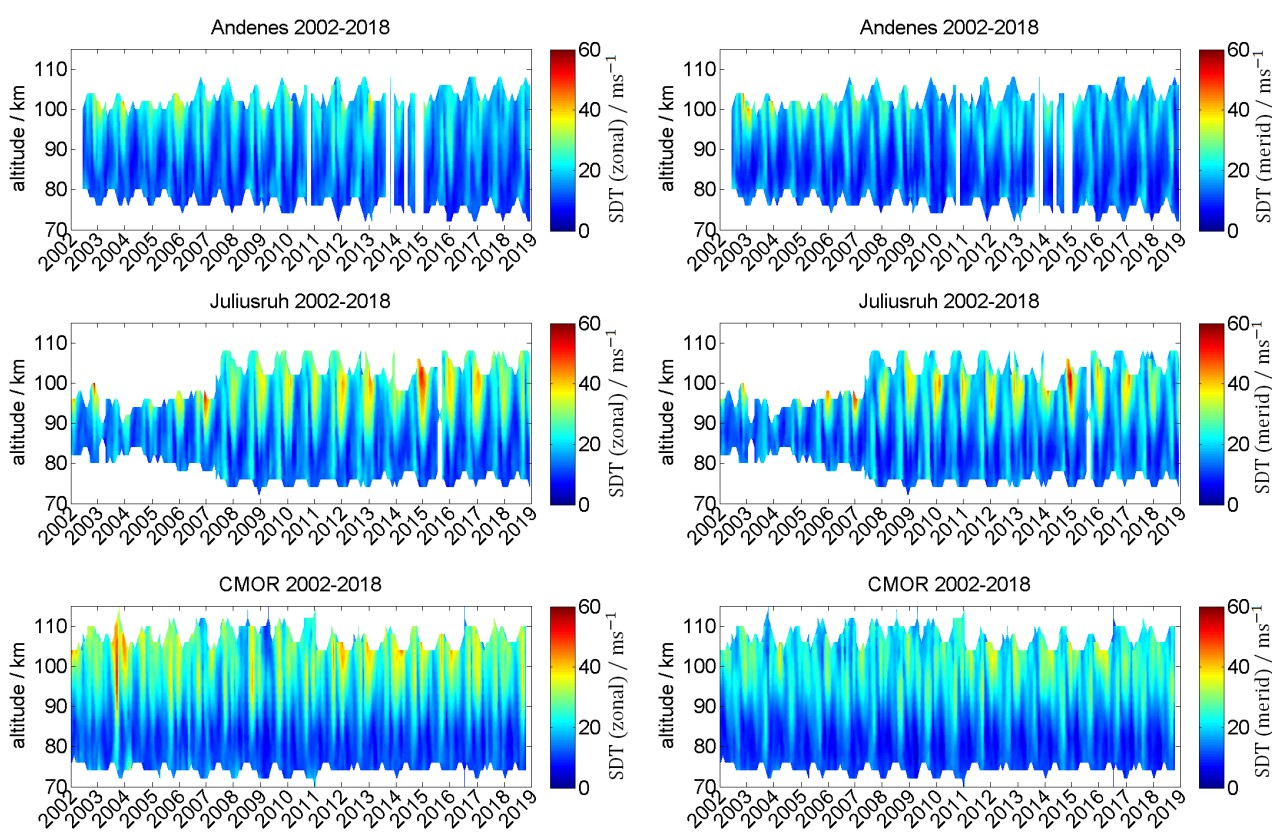

**Figure 11.** Same as Figure 7, but for the semidiurnal tidal components.

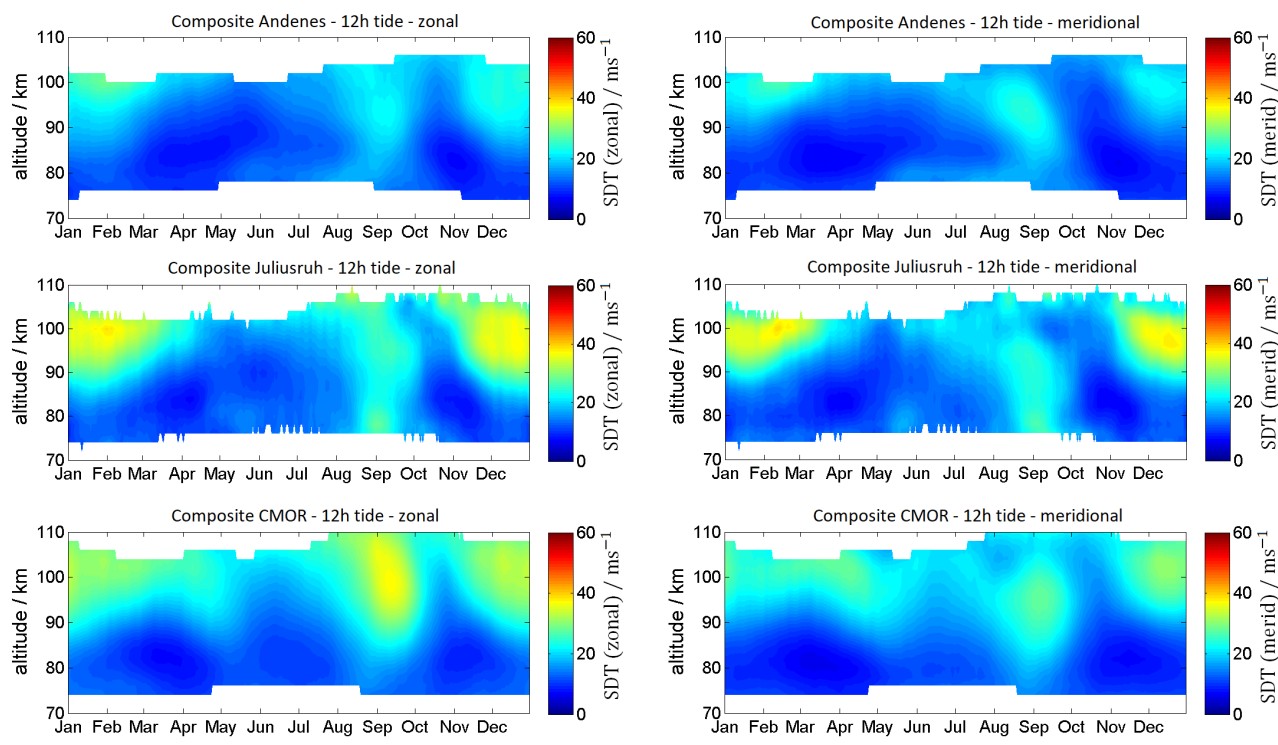

**Figure 12.** Same as Figure 8, but for the semidiurnal tidal components.

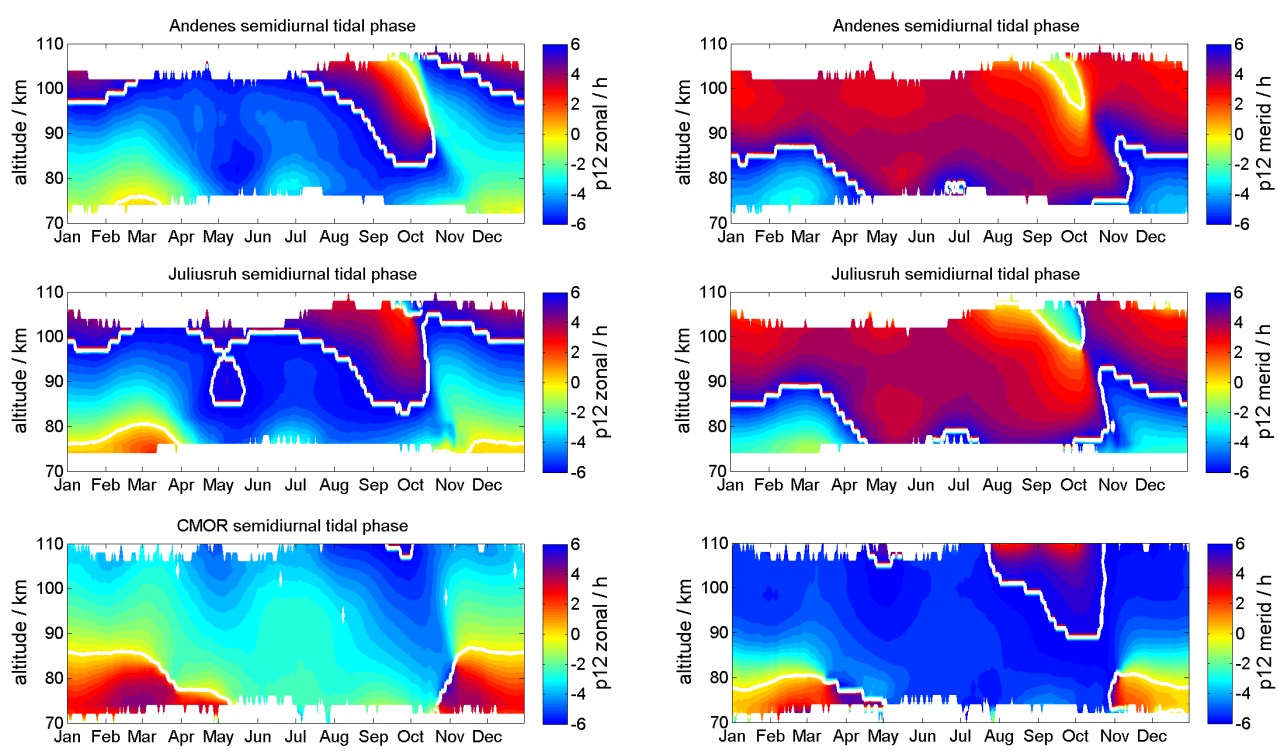

**Figure 13.** Same as Figure 9, but for the semidiurnal tidal components.

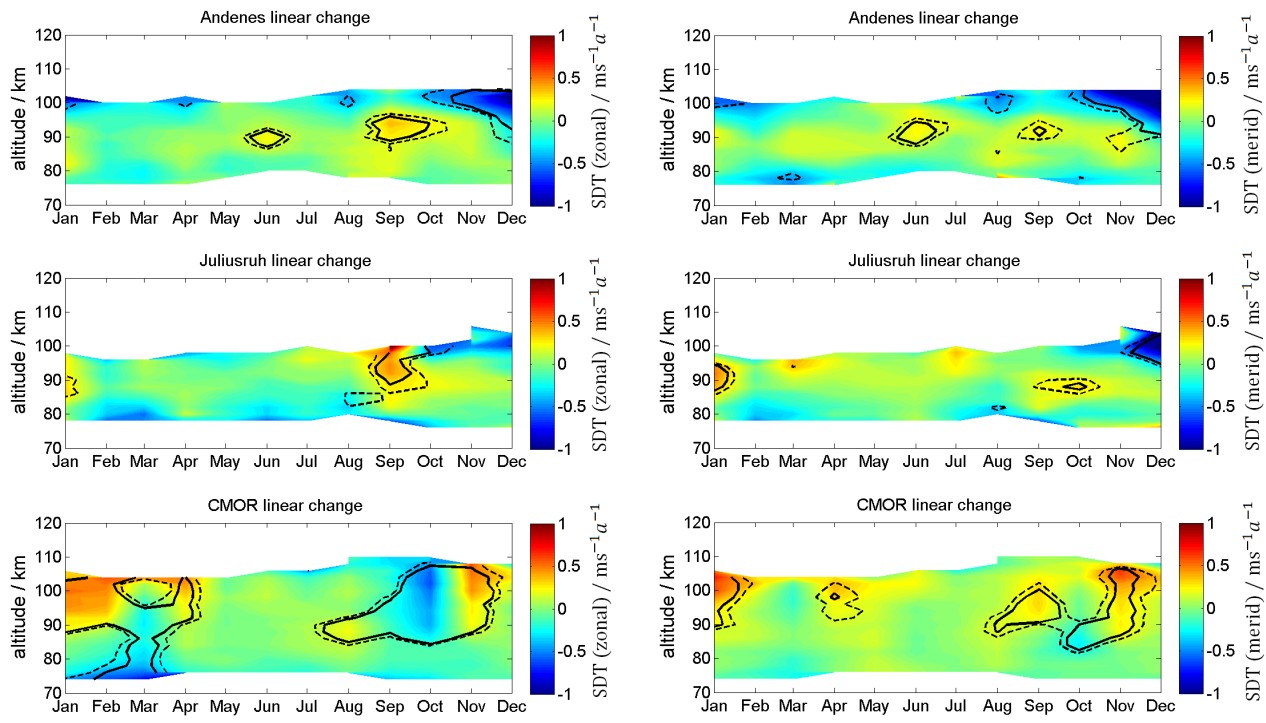

**Figure 14.** Same as Figure 10, but for the semidiurnal tidal components.

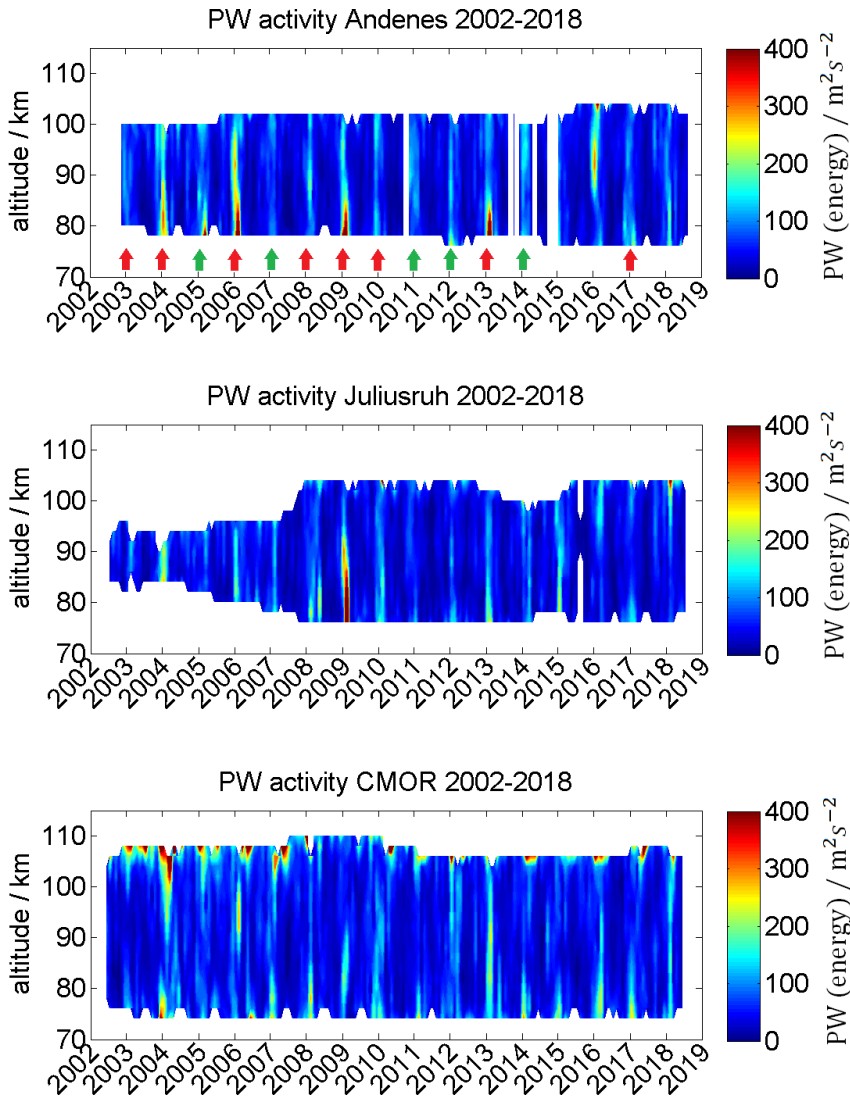

**Figure 15.** Time-series of planetary wave energy for Andenes (top), Juliusruh (middle), and CMOR (bottom). The red (green) bold arrows corresponds to winter with a major (minor) sudden stratospheric warming.

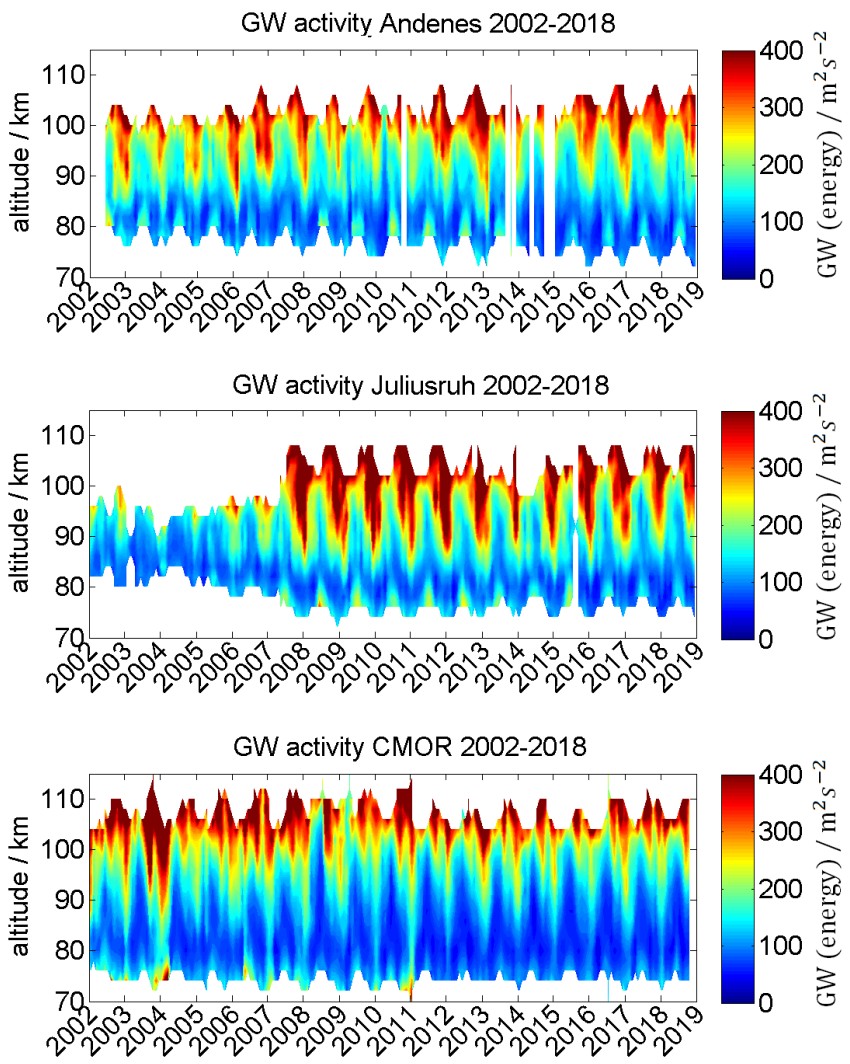

**Figure 16.** Time-series of kinetic gravity wave energy for Andenes (top), Juliusruh (middle), and CMOR (bottom).

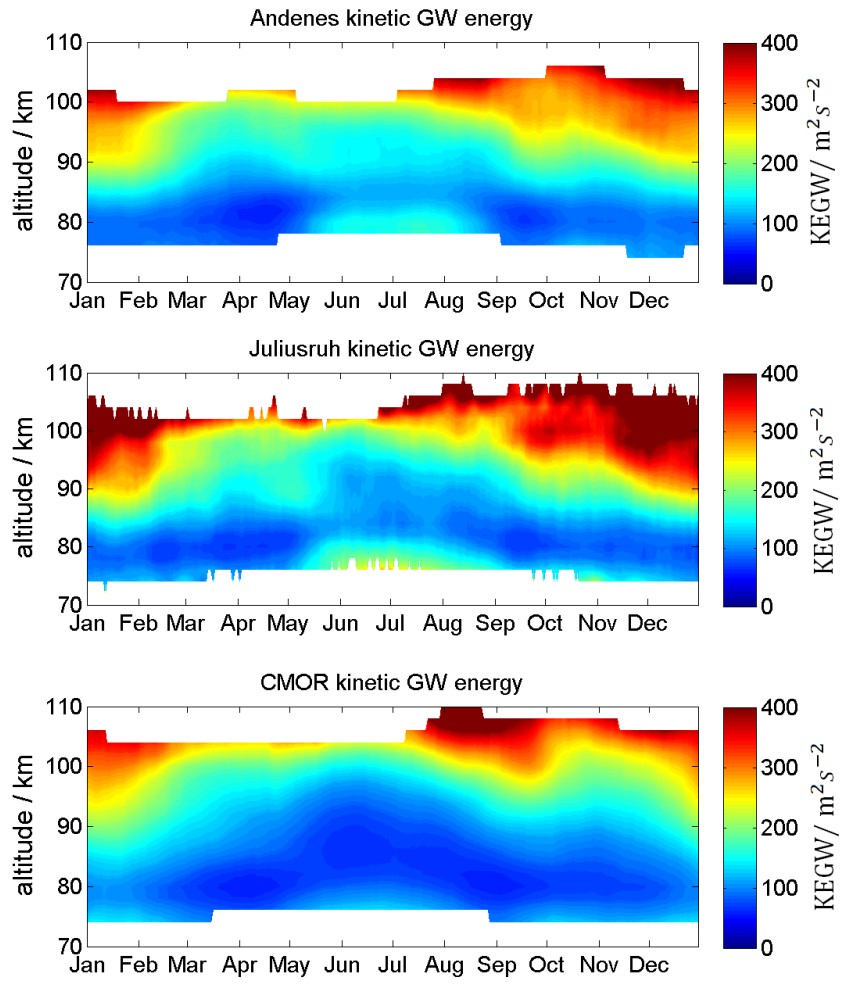

**Figure 17.** Composite of kinetic gravity wave energy for Andenes (top), Juliusruh (middle), and CMOR (bottom).

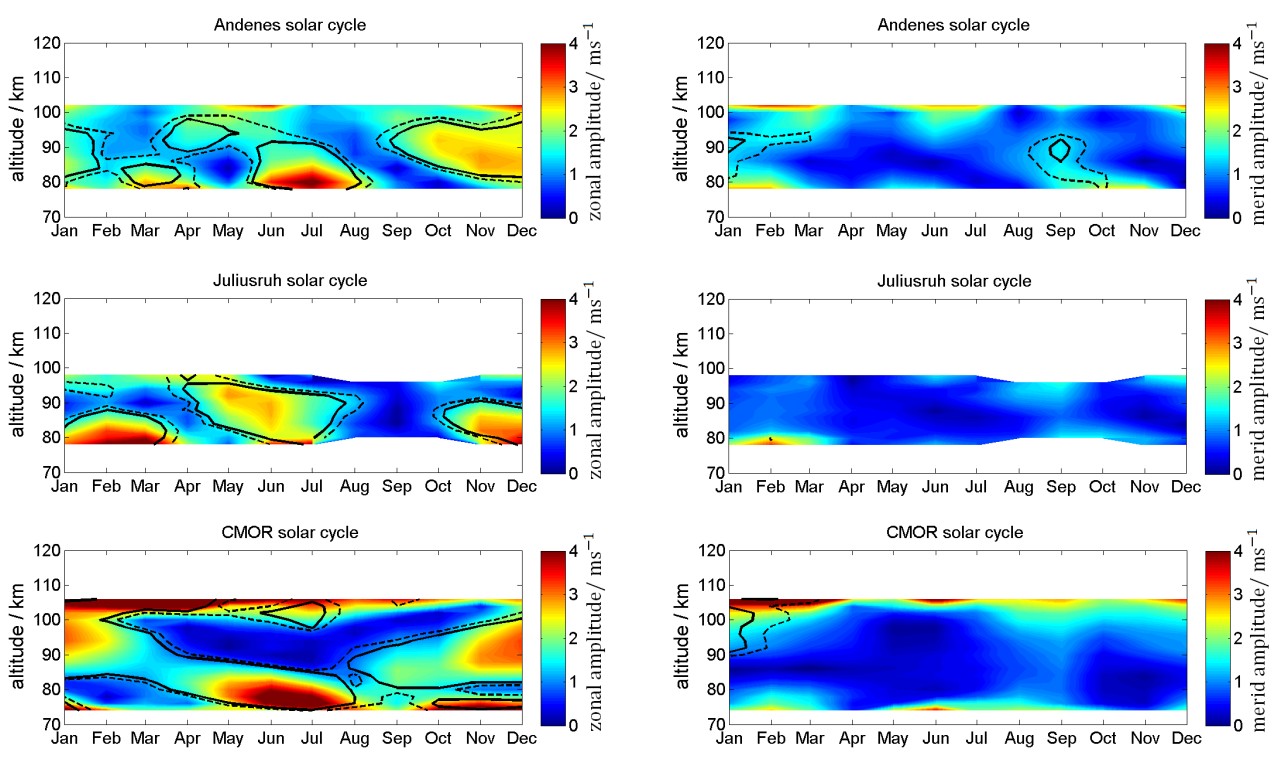

**Figure 18.** Linear change of the solar radiation on the zonal (left) and meridional (right) wind for Andenes (top), Juliusruh (middle), and CMOR (bottom). The solid black lines corresponds to 95% significance, the dashed black lines to the 90% significance.

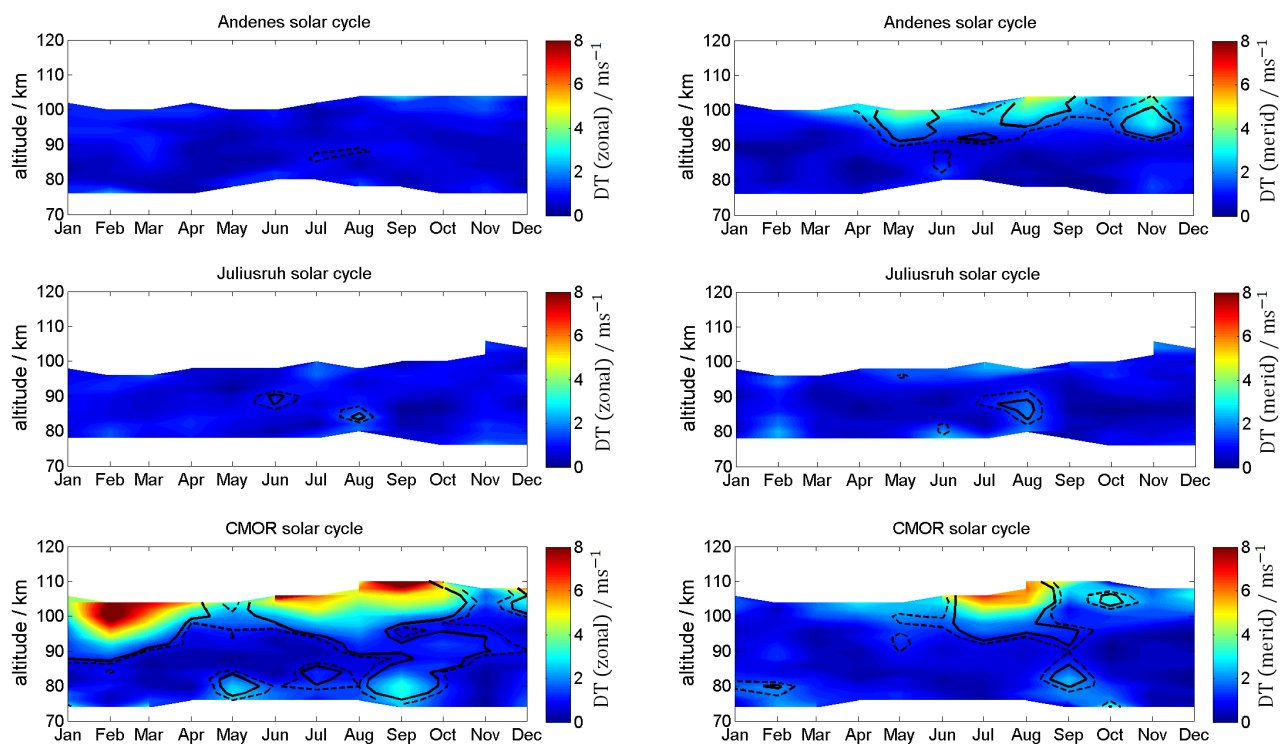

**Figure 19.** Linear change of an 11-year oscillation on the diurnal zonal (left) and meridional (right) tidal component for Andenes (top), Juliusruh (middle), and CMOR (bottom). The solid black lines corresponds to 95% significance, the dashed black lines to the 90% significance.

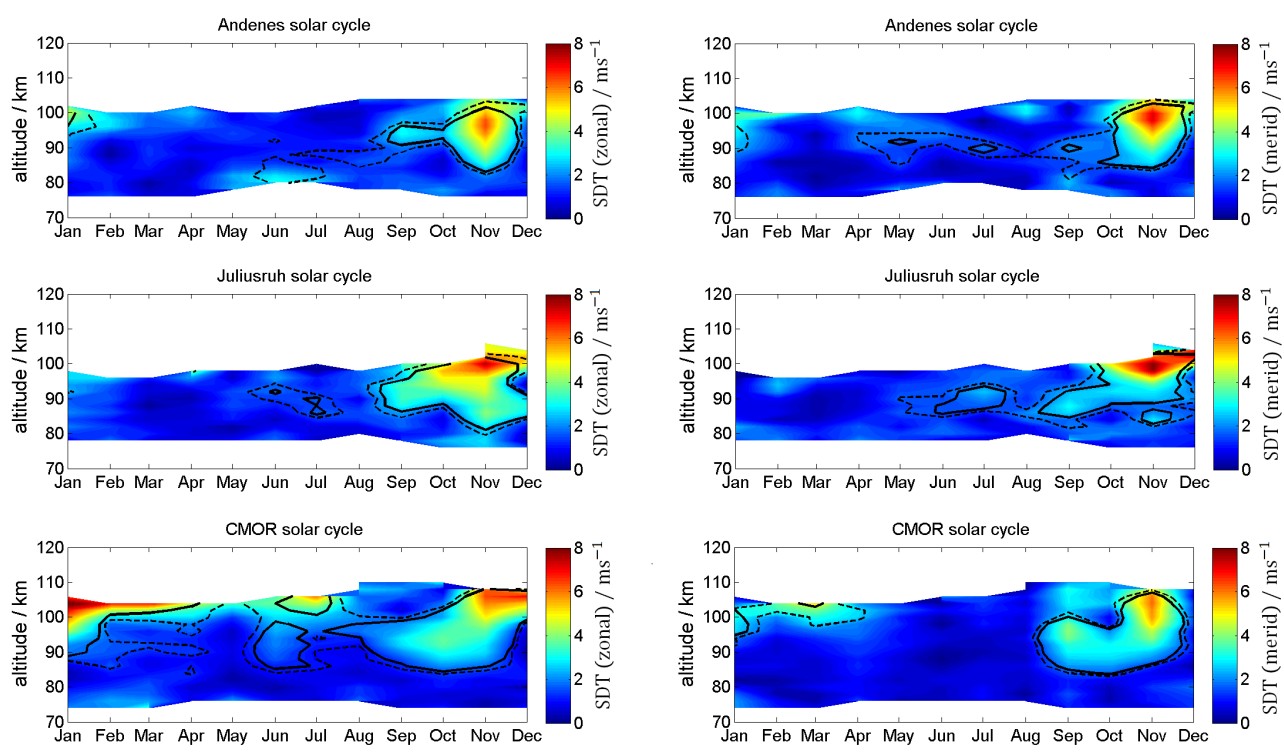

**Figure 20.** Same as Figure 19, but for the semidiurnal component.

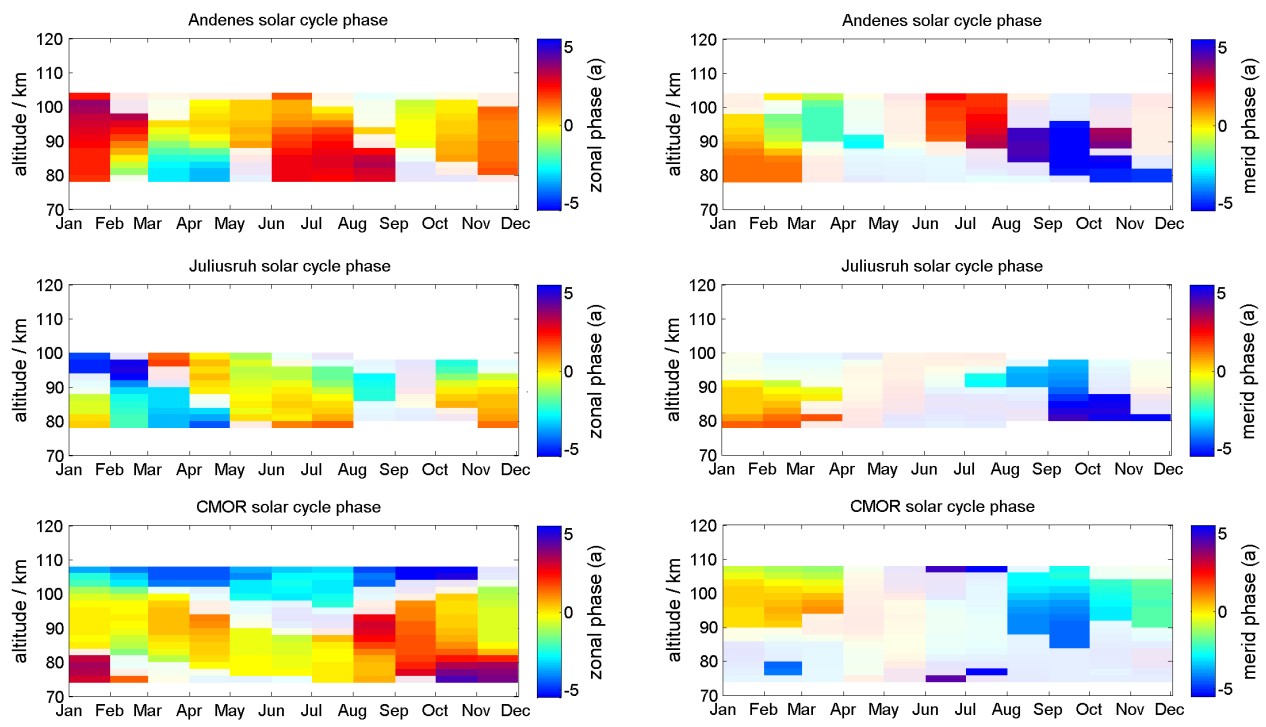

**Figure A1.** Phase information of the solar cycle fit for the wind component. The shaded areas are not significant.

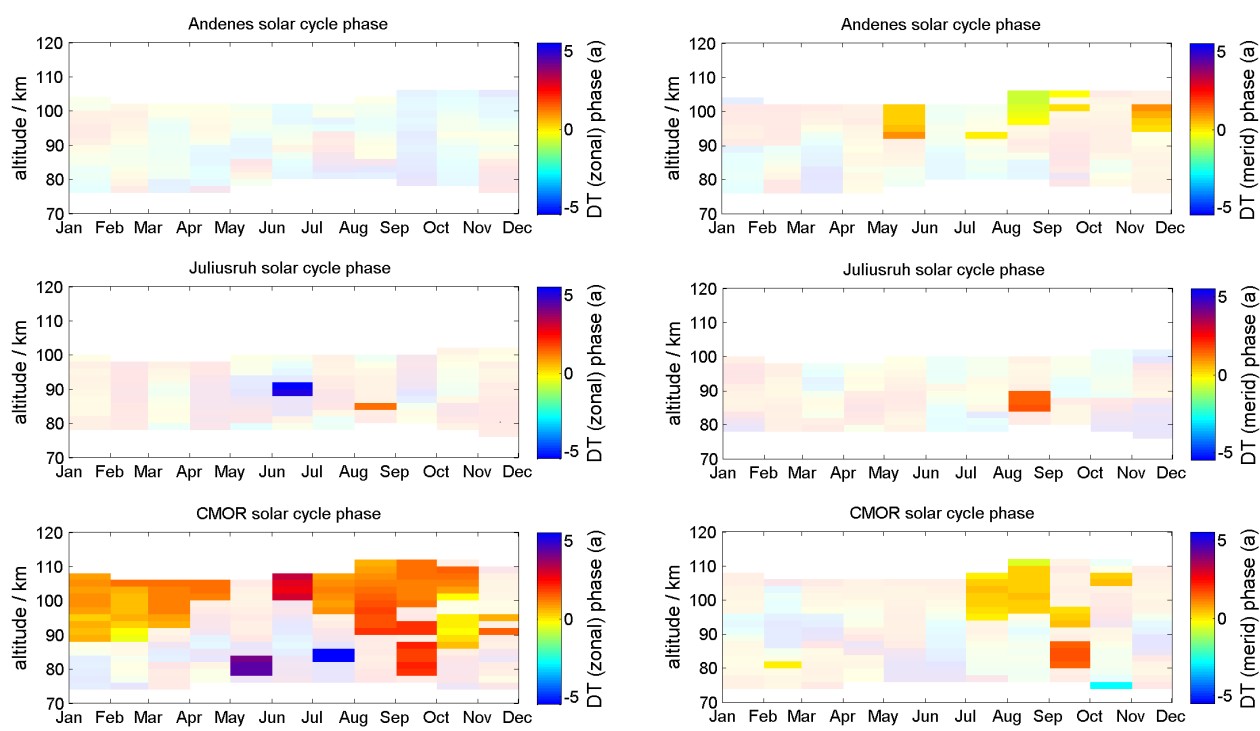

**Figure A2.** Same as Figure A1, but for the diurnal component.

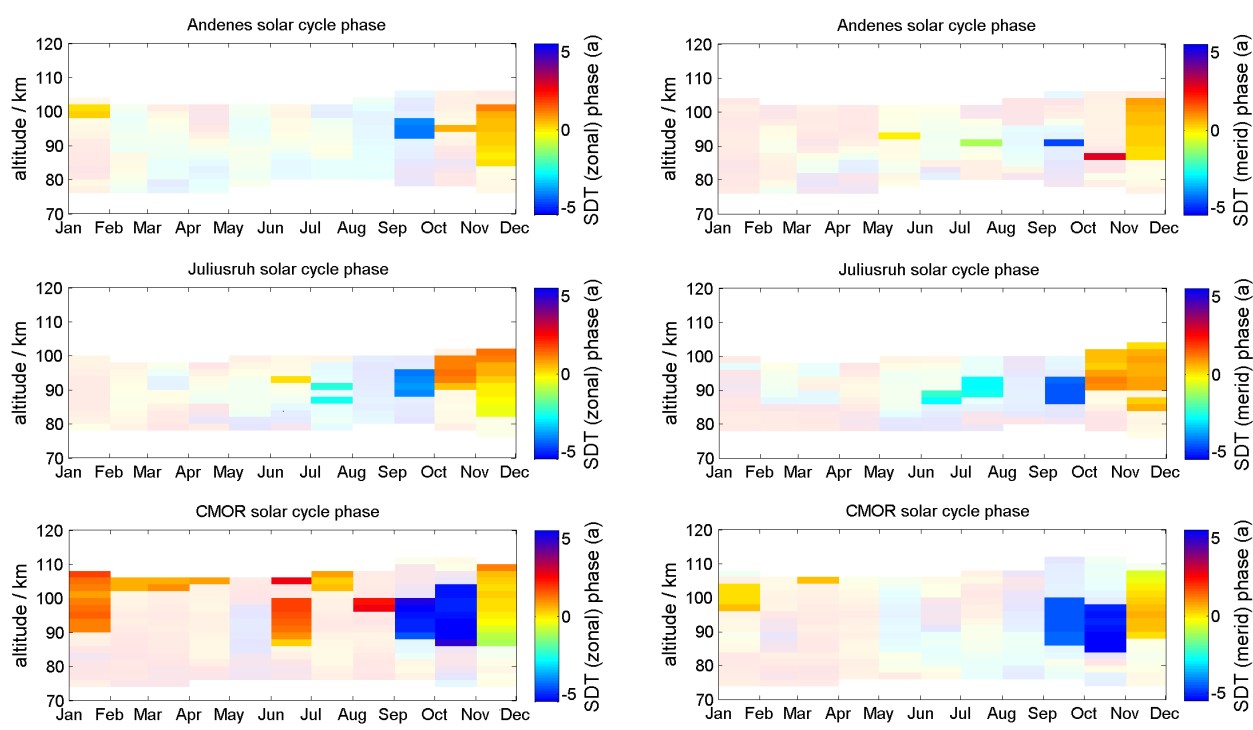

**Figure A3.** Same as Figure A1, but for the semidiurnal component.