# Peer review of "Climatologies and long-term changes of mesospheric wind and wave"

_Annales Geophysicae, 2019_

## Referee Comment (RC1) · Anonymous Referee #1 · 24 May 2019

General comments:

This manuscript describes long term (Jan 2002 to Dec 2018) measurements of MLT winds at 3 observation sites at mid and high latitudes. Out of these radar based wind measurements the authors derive climatologies of mean winds, tides, gravity and planetary waves with respect to a latitudinal dependence which makes it to a very valuable contribution. This study focused on the observational part, but for future studies, involving for example GCM simulations, it could be an interesting point of reference for comparison or validation.

An important result includes, that LTC, SDT and PW activity in the MLT region is highly

dependent on the latitude and the authors are definitely right in saying that ongoing continuous measurements at various latitudes are important in the future. The conclusions give a nice list of the broad range of achieved results. Although the structure of manuscript is very clear ( e.g. the handling of the results for the different wave types), it is a bit difficult to follow whole story with such a large number of plots. At the moment, I do not see how this could be improved for this study (and 3 observation sites) without splitting the whole manuscript. But this, I really do not want to foster, because it would also destroy the completeness of the study.

After some minor revisions (see specific comments) and technical corrections, I would recommend to publish this paper in Annales Geophysicae.

Specific comments:

- The abstract could be slightly improved. Summarize the kind of difference you observe for the DT and SDT (page 1, line 8). At the end, maybe include in one sentence what the main influence of the 11-year solar cycle is.

- Please check the use of the word "tendency" throughout the manuscript, if it is always scientifically appropriate. It may occur too often in some paragraphs (e.g. page 8).

- Page 1, line 16: For completeness, also passive microwave radiometry from ground can provide wind measurements (at least up to the upper mesosphere).

- Page 2, lines 5ff: Here I suggest to add a small paragraph on GCMs in regard to what is state of the art and in context to this study. Have you considered to explicitly compare your observations to GCM simulations?

- Page 3, lines 13-14ff: Can you give here a short explanation why the mentioned residuals are a good indicator of GW activity?

- Page 4, line 24: Regarding the composite of different radar systems, is there a homogenization applied to the different data sets? And if not, how do you ensure that the homogeneity is good enough to use it for LTC?

- Page 5, lines 5ff: Here you describe the usage of triple frequency observations for the CMOR radar. It would be nice to have more details on the homogenization technique and how you compiled the data set.

- Page 6, lines 16-18: Here I suggest to either proof that there is no phase delay/direct physical causality (with e.g. an appropriate literature study) or just say that you do not use the F10.7 proxy. But for clarification, I would prefer the first propose. Otherwise this sentence sounds a bit too speculative.

- Page 7, line 12: What is the reason of choosing a window size of 5 days in regard to obtaining the mentioned climatologies?

- Page 8, line 32: Can you give an explanation for this observation at CMOR (strongest maximum and mean amplitudes)?

- Page 11, lines 1-2: This sentence is not clearly understandable. Maybe the verb "modified" is a bit misleading here.

Technical corrections:

- Regarding the notation, if units of physical quantities are in the denominator, contain numbers, and are abbreviated, they should be formatted with negative exponents. Consider correcting this in the whole manuscript.

- Page 1, line 1: "...report on long-term..."

- Page 2, line 13: Please correct: "spatial extents"

- Page 3, line 15: Please correct: "the GW residual"

- Page 4, line 29: For those who are not familiar with a "7-bit Barker code" can you explain and/or give a reference?

- Page 7, line 15: The structure of the sentence "Only during the fall transition..." has to be improved. It does not sound correct.

- Page 8, lines 2-3: The meaning of "... and periods of a few years" is not clear to me in this sentence. Please try to improve this.

- Page 8, line 21: Maybe change the expression "seems to" to "is decreasing" in order to be more precise. Or are you unsure about this decrease. What about the uncertainty in this case?

- Page 9, line 31: Please correct: "...are indicate" to "are indicating" or only "indicate".

- Page 10, line 31: Probably there is an "and" missing in between "...during after...".

- Page 12, line 32: Correct: "MR are show" to e.g. "MR show".

- Page 13, line 9: Delete "the" after "Although".

- Figure 6: There is space to increase the label sizes for better readability.

---

## Referee Comment (RC2) · Anonymous Referee #2 · 11 Jun 2019

The authors present long-term analyses of mean winds, tides, and PW and GW proxies obtained from meteor radar observations at three sites at middle and high latitudes since 2002. They apply a new method to derive the tides. Long-term changes (linear trends) are also presented, and a brief analysis of possible solar cycle effects on the winds.

The results are certainly interesting and worth to be published. However, description and presentation are partly unclear, and some revision is necessary. Specific comments are given below.

page 2, l 14-20: In the presentation, stationary and traveling PWs are confused. Land-sea differences mainly force SPW, but traveling PW can be formed by other mechanisms. Energy transport is mainly due to breaking waves. And of course SPW cannot be seen by single radars.

Page 3, l 14-15: "To estimate the PWs…" This sentence is rather confusing here. GW are calculated as residuals from the tidal fit (during one day?). Please state this clearly.
PW are described later, remove this here.

Page 3: l 18-25: This paragraph comes somewhat unmotivated here. I should be moved up, e.g. before the discussion of Figure 1.

Page 5, l 19: did you apply a fit with 12 and 24 hrs, or with 3 components according to Eq. 1 and just did not show the terdiurnal component? Please clarify.

Page 6. l 14-16: This is unclear to me. Phase delay can be analyzed with lagged correlation, and I also do not see why a 11-year fit provides more physical insight than a regression.

Page 7, l 1: Gaussian error distribution. The degree of freedom decreased if there is an autocorrelation, and this should be the case e.g. through the solar cycle. Please comment on this.

Page 8, l 5, Figure 5: The presentation of significance levels is unclear. There are frequent trend changes from positive to negative, and therefore zero trends, which are nevertheless indicated as significant. So what is the test hypothesis?

Section 4.1, 1st paragraph: it may be worth to note that the seasonal cycle of the DT differs from the one presented in earlier papers like Portnyagin et al., 2004 or Jacobi 2012 in particular at lower heights, e.g., there is no or only weak spring/autumn enhancement. Earlier papers frequently used longer time windows for analysis. Differences then may be explained by strong day-to-day phase changes especially during summer. I would be interested in knowing whether this is the case.

Page 9, l 31, see also discussion and conclusion. Circular polarization means that u and v have the same amplitude and 90° phase difference. The latter is not shown.

Page 10, l 10. SSW are connected with mean wind changes at the time scale of days or few weeks, i.e. in the period range of PWs. So how do you distinguish between them? This also questions your statement on lines 11/12 that PWs are less strong during minor warmings.

Section 5: Earlier papers have shown that a possible solar cycle effect can be in phase, out of phase, or lagged with respect to the solar cycle. So here the phase should be provided to be able to compare with literature results.

Minor issues

Abstract, l7: it is more common to handle the zonal and meridional wind components separately: ... the south and to the west.

Abstract l 10: The influence of the 11-year solar cycle on the winds and tides is presented. Please provide the main result here.

Page 2, l 27: with the exception of… do you mean „according to" ?

Page 2, l 34: wavesrestoring → wave´s restoring

Page 3, l 12: I assume that T3 = 8 hrs, please add this information

Page 3, l 13: PW → GW

Page 3, l 16: nearly two weeks → ten days

Page 3, l 28: Stober et al. 2012 does not deal with LTC

Page 4, l 2: Portnyagin et al., 2004 did not investigate LTC, but shows a long-term climatology

Page 4, l 3: Lukianova et al., 2018 did not investigate LTC, but interannual variability, in particular SSW

Page 4, l 5: meridional wind trends by Jacobi et al., 2015 differ between seasons after 2005

Page 5, l 27: benefit is → benefit of

Page6:, l 4: seasonal → annual

Page 7, l 25: Jacobi et al. 2015 only showed data at ~90 km, do you mean Jacobi, 2012?

Page 7. l 35: 2006 → 2007

Page 8, 2nd paragraph: The description is confusing. Please state clearly to which month and which altitude you refer to.

Figure 12: again, there are zonal trends indicated as significant.

Reference Portnyagin et al., 2004: provide doi: https://doi.org/10.5194/angeo-22-3395-2004

---

## Author Comment (AC1) · 31 Jul 2019

General reply: We thank the referee for the recognition of the work and for constructive suggestions and comments that help to improve the paper. This paper focuses on observations and it is intended to provide a summary and diagnostic of MR measure-

ments as reference for other observations and also GCM models. A detailed investigation of the possible causes for all the effects is beyond the scope of the paper and has to be done in additional works. The revised version can be found in the supplements.

Specific comments:

- The abstract could be slightly improved. Summarize the kind of difference you observe for the DT and SDT (page 1, line 8). At the end, maybe include in one sentence what the main influence of the 11-year solar cycle is.

Reply: Thank you for the comment.

We added to the abstract the following part: The diurnal tides show nearly no significant long-term changes, while changes for the semidiurnal tides differ regarding altitude. Andenes shows only during winter a tidal weakening above 90 km, while for CMOR occur an enhancement of the semidiurnal tides during the winter and a weakening during fall. [. . .] The influence of the 11-year solar cycle on the winds and tides is presented. The mean winds exhibit a significant amplitude response, for the zonal component below 82 km during summer and from November - December between 84 and 95 km at Andenes and CMOR. The SDT show a clear 11-year response at all locations, from October to November.

- Please check the use of the word "tendency" throughout the manuscript, if it is always scientifically appropriate. It may occur too often in some paragraphs (e.g. page 8).

Reply: Thanks for the comment. We checked and reformulated sometimes the word "tendency". In the manuscript we avoid to use the term "trend", which would be an often used alternative, simply because for a trend requires at least 30 years of data.

- Page 1, line 16: For completeness, also passive microwave radiometry from ground can provide wind measurements (at least up to the upper mesosphere).

Reply: We added the passive microwave radiometer to the list.

[Figure]

- Page 2, lines 5ff: Here I suggest to add a small paragraph on GCMs in regard to what is state of the art and in context to this study. Have you considered to explicitly compare your observations to GCM simulations?

Reply: Within the current study we don't want to compare our observations and the corresponding filter approach to GCM simulations, but we are planning to do a comparison between our findings and GCM simulations. Furthermore, we are planning to applicate the adaptive spectral filter approach on other instruments and GCM data.

We added a small part regarding GCM-state of the art: Basically, climatologies of winds and tides in the mesosphere are well represented in GCMs. With the onset of the mesopause, differences occur between models and observations, which are shown in several studies. Yuan et al. (2008) showed differences between 3 models and observations, as well as, also between models itself, by mentioning that the height of the summer mesopause differs. Stronger differences occur during the winter, opposite prevailing wind directions occur above the mesopause between models and observations Pokhotelov et al. (2018). A reason for these differences is probably based in using different gravity waves parameterizations.

- Page 3, lines 13-14ff: Can you give here a short explanation why the mentioned residuals are a good indicator of GW activity?

Reply: We did look at the total spectrum of fluctuations and identified the different wave types and put notch filters to wave with well-defined periods such as tides. The 8-hour tide is usually considered as most challenging to be extracted and contains also a significant amount of GW activity. This is also the reason why we do not provide a climatology for this tide in the manuscript.

We added to the manuscript: The gravity wave activity is the residuum, which includes all fluctuations different than tides and planetary waves.

- Page 4, line 24: Regarding the composite of different radar systems, is there a ho-

mogenization applied to the different data sets? And if not, how do you ensure that the homogeneity is good enough to use it for LTC?

Reply: We thank the reviewer for making this comment. The quality of the meteor radar measurements depend on the quality of the phase calibration to determine the angle of arrival, the accuracy to estimate the Doppler velocity from the ambipolar diffusing meteor trails and the range calibration of the system. However, the most crucial problem could be a phase drifting of the interferometer, thus, the phase calibration and phase stability monitoring is the most critical part.

Range calibration: The Juliusruh and Andenes MR were frequently range and power calibrated using a delay line feeding an attenuated transmitter pulse into the receivers and measuring a well-defined delay. The CMOR radar is cross calibrated by comparing optical trajectory solutions from camera systems with the individual radar meteors. Further, it is possible to use the time of flight solution without any range measurement as independent validation.

Doppler measurements: The Doppler measurement is less prone to measurement errors compared to the other two potential error sources as we use always the same software to estimate the radial velocities. Potential degradations in the local oscillators are compensated as the doppler measurement of the IP and QP voltages is always generated from the signal that was used in the transmission signal synthesis.

Phase calibration: The standard SKiYMET software runs, on a user defined frequency, an internal relative phase check to identify a possible degradation between the different receiver channels of the interferometer. Further, the Juliusruh and Andenes MR were 2 times per year maintained and all antennas/cables were controlled for potential phase mismatches and corrected. Further, we track the position for the major meteor showers throughout the year between all systems and check for potential mismatches.

The CMOR radar is also well-maintained and provides a detailed log file. In addition, the CMOR data can be cross validated by the other three frequencies and the optical

meteor observations. The CMOR meteor shower catalogue is also cross validated to optical observations.

- Page 5, lines 5ff: Here you describe the usage of triple frequency observations for the CMOR radar. It would be nice to have more details on the homogenization technique and how you compiled the data set.

Reply: Homogenization of multiple frequency datasets (Juliusruh 32.55 MHz and 53.5 Mhz, CMOR 17 MHz, 29 Mhz and 38 Mhz) are compiled on the basis of individual specular meteor trail measurements. The SKiYCORR software collects and reduces the data separately and provide a measurement summary containing the information for each individual meteor measurement such as radial velocity, radial velocity error and angle or arrival, time of detection and some other parameters.

Each radar provides an independent measurement for each meteor, although it is possible that the same meteor is detected at the different frequencies. All quality-controlled meteor detections are weighted by their statistical uncertainty and enter the wind analysis described in Stober et al., 2018.

Before we compile the multi-frequency combined wind data sets we run cross validations for each frequency and compare and cross-validate all systems for consistency. This is done for the CMOR data set as well as for the Juliusruh data set as long as multiple frequency observations were available. The Andenes MR is cross-validated and compared on a campaign basis to other Scandinavian meteor radars.

We added a subsection called "Homogenization of time series" to the manuscript: (for the last two remarks): The instruments used in this study were operational for almost two decades and some meteor radars did undergo substantial maintenance and modifications on the hardware. Most crucial for the wind measurements are the phase calibration and stability, the range sampling and the Doppler measurement.

The Andenes and Juliusruh meteor radar were maintained twice a year including a

test of the phase match of the cables and antennas. Further, the SKiYCORR software runs a phase test and provides a summary file of the impedance for each channel and day indicating potential problems. In addition, to the regular maintenance the CMOR meteor radar interferometry (phases) is cross validated to optical observations. In particular, meteor showers are monitored with CMOR throughout the year providing another source of information on the phase stability. The Andenes and Juliusruh meteor radar were also checked and cross validated using selected meteor showers during the course of the year.

Both European meteor radars were frequently range and power calibrated using a delay line (Latteck et al. 2008, Stober et al. 2010). The CMOR radar is also routinely checked for potential issues in the range sampling applying various cross calibrations. All systems used the same software package over the complete time span to derive the Doppler velocities to avoid artefacts due to changes in the parameter estimation (e.g., Doppler velocity or the velocity uncertainty).

Before the multi-frequency data sets for CMOR and Juliusruh are compiled, we analyze the winds for each frequency independently and cross-validate the resultant time series. If one instrument shows systematic issues in the wind time series compared to the other instruments and the climatology, this data is flagged and no longer considered in the finally compiled and merged wind time series. The Andenes meteor radar data is campaign wise cross-validated with other meteor radars in Norway.

References : Latteck, R., Singer, W., Morris, R., J., Hocking, W., K., Murphy, D. J., Holdsworth, D., A., and Swarnalingam, N.: Similarities and differences in polar mesosphere summer echoes observed in the Arctic and Antarctica, Annales Geophysicae, 26, https://doi.org/10.5194/angeo-26- 2795-2008, www.ann-geophys.net/26/2795/2008/, 2008.

Stober, G., Jacobi, C., and Keuer, D.: Distortion of meteor count rates due to cosmic radio noise and atmospheric particularities, Advances in Radio Science, 8, 237–

241, https://doi.org/doi:10.5194/ars-8-237-2010, https://doi.org/10.5194%2Fars-8-237-2010, 2010.

- Page 6, lines 16-18: Here I suggest to either proof that there is no phase delay/direct physical causality (with e.g. an appropriate literature study) or just say that you do not use the F10.7 proxy. But for clarification, I would prefer the first propose. Otherwise this sentence sounds a bit too speculative.

Reply: We added some literature pointing out the complexity and diversity of the coupling and correlation between solar proxies and MLT winds. It appears to be a very controversial issue. However, we also looked into the possibility of potential phase delays between solar activity (F10.7 or sunspot number) and geomagnetic indices and solar wind. Kilcik et al., 2017 computed potential phase delays between the different indices. This delays could be months to years. However, it is beyond this study to investigate in detail, which may would get the highest correlation in the regression model. Further, we now added all phase information to the climatologies and the solar cycle forcing with season. This was also recommended by the second reviewer.

We modified the text into: The LTC and solar cycle effect are derived by using a linear trend model plus an 11-year oscillation, which is not tied to the F10.7 solar radio flux or the sunspot number. Qian et al. (2019) analyzed WACCM-X and wind observations above Collm (51°N, 13°E) and found that the wind signature is less statistical significant than the temperature response to the solar radio flux. Other studies exploring the stratospheric/tropospheric response to solar forcing indicate a more clear dependence (Rind et al., 2008; Salby and Callaghan, 2006; Lu et al., 2017) on solar activity. At the MLT, the wind seems to be less directly influenced by the F10.7 or sunspot number. Pokhotelov et al. (2018) found almost no correlation between the occurrence of mesospheric echoes at mid-latitudes and the solar radio flux (F10.7), but a clear dependence on the occurrence of these echoes due to meridional winds. Further, Stober et al. (2014) investigated the neutral air density response during the solar cycle 23 and found a phase delay of almost 1 year between the F10.7 proxy and the neutral air

density variation. [. . .]

Qian, L., Jacobi, C., and McInerney, J.: Trends and Solar Irradiance Effects in the Mesosphere, Journal of Geophysical Research: Space Physics, 124, 1343–1360, https://doi.org/10.1029/2018JA026367, https://agupubs.onlinelibrary.wiley.com/doi/abs/10.1029/ 2018JA026367, 2019.

Rind, D., Lean, J., Lerner, J., Lonergan, P., and Leboissetier, A.: Exploring the stratospheric/tropospheric response to solar forcing, J. Geophys. Res., 113, D24 103, https://doi.org/10.1029/2008JD010114, 2008.

Salby, M. L. and Callaghan, P. F.: Influence of the Solar Cycle on the General Circulation of the Stratosphere and Upper Troposphere, Space Science Reviews, 125, 287–303, https://doi.org/10.1007/s11214-006-9064-3, https://doi.org/10.1007/s11214-006-9064-3, 2006.

Lu, H., Gray, L. J., White, I. P., and Bracegirdle, T. J.: Stratospheric Response to the 11-Yr Solar Cycle: Breaking Planetary Waves, Internal Reflection, and Resonance, Journal of Climate, 30, 7169–7190, https://doi.org/10.1175/JCLI-D-17-0023.1, https://doi.org/10.1175/ JCLI-D-17-0023.1, 2017.

- Page 7, line 12: What is the reason of choosing a window size of 5 days in regard to obtaining the mentioned climatologies?

Reply: The 5 days are a mistake, we actually applied a 30 day window centered at the appropriate day to make our climatologies comparable to monthly means, which are often used in other studies.

We removed this part to avoid misunderstandings.

- Page 8, line 32: Can you give an explanation for this observation at CMOR (strongest maximum and mean amplitudes)?

Reply: Currently, we cannot given an explanation why such a strong enhancement of

the diurnal component takes place at CMOR, which further only occurs in the zonal component, and mainly during the winter. Additionally in Figure 7, an enhancement in the years 2002-2005 can be observed. Furthermore, this enhancement does not take place in the dominant semidiurnal component. Probably in a future work, with the help of GCMs, we are going to investigate why the zonal diurnal component results in such strong maximum and mean amplitudes.

- Page 11, lines 1-2: This sentence is not clearly understandable. Maybe the verb "modified" is a bit misleading here.

Reply: Thank you for the comment.

We changed the phrase into: The amplitudes of semidiurnal tidal components range between 3 and 6 m/s.

Technical corrections: - Regarding the notation, if units of physical quantities are in the denominator, contain numbers, and are abbreviated, they should be formatted with negative exponents. Consider correcting this in the whole manuscript.

General reply: We appreciate and adopted the comments regarding the technical corrections.

- Page 1, line 1: "...report on long-term..." - Reply: we corrected the word

- Page 2, line 13: Please correct: "spatial extents" - Reply: we corrected the word

- Page 3, line 15: Please correct: "the GW residual" - Reply: we corrected the word

- Page 4, line 29: For those who are not familiar with a "7-bit Barker code" can you explain and/or give a reference? -Reply: we added a reference (Hocking et al., 2016)

- Page 7, line 15: The structure of the sentence "Only during the fall transition..." has to be improved. It does not sound correct. -Reply: we changed the sentence into: During the fall transition, Juliusruh show for a month at altitudes above 95 km westward directed wind.

- Page 8, lines 2-3: The meaning of "... and periods of a few years" is not clear to me in this sentence. Please try to improve this. -Reply: we removed the sentence.

- Page 8, line 21: Maybe change the expression "seems to" to "is decreasing" in order to be more precise. Or are you unsure about this decrease. What about the uncertainty in this case? -Reply: we changed the phrase into "is decreasing"

- Page 9, line 31: Please correct: "...are indicate" to "are indicating" or only "indicate". - Reply: we corrected phrase into "indicate"

- Page 10, line 31: Probably there is an "and" missing in between "...during after...". - Reply: corrected

- Page 12, line 32: Correct: "MR are show" to e.g. "MR show". - Reply: thanks for the comment

- Page 13, line 9: Delete "the" after "Although". - Reply: thanks for the comment

- Figure 6: There is space to increase the label sizes for better readability -Reply: we increased the font size

Please also note the supplement to this comment:
https://www.ann-geophys-discuss.net/angeo-2019-51/angeo-2019-51-AC1-supplement.pdf

―――――――――――――――――――――

**Supplement:**

[revised manuscript text omitted]

---

## Author Comment (AC2) · 31 Jul 2019

General reply: We thank the referee for the comments and appreciate the work which was done referee to increase the quality of the manuscript. We reply to all the raised points and also followed the recommendation to show the phases of the tides as well as

for the solar cycle effects. This requires rather substantial text edits in some parts of the manuscript. We appreciate this reviewer comments as it makes this work a much more complete reference on long term MR observations and climatologies of atmospheric waves.

A revised version of the manuscript can be found in the appendix.

Specific comments:

page 2, l 14-20: In the presentation, stationary and traveling PWs are confused. Land-sea differences mainly force SPW, but traveling PW can be formed by other mechanisms. Energy transport is mainly due to breaking waves. And of course SPW cannot be seen by single radars. Reply: Thanks for the comment.

We reformulated the part: Large scale PWs are primary formed in the troposphere by topography and diabatic heating. They influence the general circulation by transferring warm air from the tropics to the poles and by returning cold air towards the tropics. Planetary waves with periods of 2, 5, 10, and 16 days and their role in dynamical processes within the MLT and regions above and below have been frequently discussed in literature (e.g., Iimura et al., 2015; Egito et al., 2016; Matthias and Ern, 2018).

Page 3, l 14-15: "To estimate the PWs..."This sentence is rather confusing here. GW are calculated as residuals from the tidal fit. Please state this clearly. PW are described later, remove this here.

We modified the text: The gravity wave activity is the residuum, which includes all fluctuations different than tides and planetary waves.

The estimation of the planetary waves follows in the section "Adaptive spectral filtering of time series".

Page 3: l 18-25: This paragraph comes somewhat unmotivated here. It should be moved up, e.g. before the discussion of Figure 1.

Reply: Thank you for the comment. We moved the part before the discussion of Figure 1.

Page 5, l 19: did you apply a fit with 12 and 24 hrs, or with 3 components according to Eq.1 and just did not show the terdiurnal component? Please clarify.

Reply: We apply the fit for all three tidal components, but within this study the gravity wave residuum includes the terdiurnal component.

We added to the text: The ASF provides a wave decomposition of our original observed time series into a daily mean wind, a diurnal and semidiurnal tide as well as a gravity wave residuum with an hourly resolution. Here, the gravity waves residuum also includes the terdiurnal tidal component.

Page 6. l 14-16: This is unclear to me. Phase delay can be analyzed with lagged correlation, and I also do not see why an 11-year fit provides more physical insight than a regression.

Reply: We rephrased this paragraph as this sentence could be misunderstood. The second reviewer also made a comment on this paragraph. However, we agree that a lagged correlation in a multi-regression analysis would work similar than the sinusoidal fitting approach. The main reason for introducing a time or phase delay by using solar proxies is given in Kilcik et al., 2017, who investigated the time lagging between different solar and geomagnetic proxies. They identified some phase delays between F10.7 and the geomagnetic indices and the solar wind. However, such a detailed analysis is beyond the scope of the submitted paper, but maybe worth for a future investigation.

We modified the text into: The LTC and solar cycle effect are derived by using a linear trend model plus an 11-year oscillation, which is not tied to the F10.7 solar radio flux or the sunspot number. Qian et al. (2019) analyzed WACCM-X and wind observations above Collm (51°N, 13°E) and found that the wind signature is less statistical significant than the temperature response to the solar radio flux. Other studies exploring

the stratospheric/tropospheric response to solar forcings indicate a more clear dependence (Rind et al., 2008; Salby and Callaghan, 2006; Lu et al., 2017) on solar activity. At the MLT, the wind seems to be less directly influenced by the F10.7 or sunspot number. Pokhotelov et al. (2018) found almost no correlation between the occurrence of mesospheric echoes at mid-latitudes and the solar radio flux (F10.7), but a clear dependence on the occurrence of these echoes due to meridional winds. Further, Stober et al. (2014) investigated the neutral air density response during the solar cycle 23 and found a phase delay of almost 1 year between the F10.7 proxy and the neutral air density variation.

Qian, L., Jacobi, C., and McInerney, J.: Trends and Solar Irradiance Effects in the Mesosphere, Journal of Geophysical Research: Space Physics, 124, 1343–1360, https://doi.org/10.1029/2018JA026367, https://agupubs.onlinelibrary.wiley.com/doi/abs/10.1029/ 2018JA026367, 2019.

Rind, D., Lean, J., Lerner, J., Lonergan, P., and Leboissetier, A.: Exploring the stratospheric/tropospheric response to solar forcing, J. Geophys. Res., 113, D24 103, https://doi.org/10.1029/2008JD010114, 2008.

Salby, M. L. and Callaghan, P. F.: Influence of the Solar Cycle on the General Circulation of the Stratosphere and Upper Troposphere, Space Science Reviews, 125, 287–303, https://doi.org/10.1007/s11214-006-9064-3, https://doi.org/10.1007/s11214-006-9064-3, 2006.

Lu, H., Gray, L. J., White, I. P., and Bracegirdle, T. J.: Stratospheric Response to the 11-Yr Solar Cycle: Breaking Planetary Waves, Internal Reflection, and Resonance, Journal of Climate, 30, 7169–7190, https://doi.org/10.1175/JCLI-D-17-0023.1, https://doi.org/10.1175/ JCLI-D-17-0023.1, 2017.

Page 7, l 1: Gaussian error distribution. The degree of freedom decreased if there is an autocorrelation, and this should be the case e.g. through the solar cycle. Please comment on this.

Reply: We agree with the reviewer about a potential autocorrelation through the solar cycle. The applied least square solution or ordinary least squares (OLS) estimator provides only an unbiased solution, if the Gauss-Markov theorem holds. This requires that our linear regression model has uncorrelated errors and equal variances. Due to the full error propagation every month is dominated by its measurement statistics (number of detected meteors, statistical errors of radial velocities and their spatial distribution). In our linear regression model all atmospheric oscillations/patter shorter than the solar cycle and longer than a year are treated as random noise with respect to the Gauss - Markov theorem. Although our statistics is small, we are confident that the residuals are uncorrelated and, thus, our fits are close to the OLS solution.

We added the following part: We tested our confidence intervals whether they are significant by introducing two null hypothesis. The long term linear changes are tested under the assumption that there is no linear change as null hypothesis, the solar cycle is tested with the null hypothesis that there is no significant solar cycle. However, it is also important to note that there could be a potential autocorrelation in our time series due to the solar cycle. We estimated all confidence levels under the assumptions that the Gauss-Markov theorem holds and our least square estimators are an unbiased solution, viz. that the fit residuals are uncorrelated.

Page 8, l 5, Figure 5: The presentation of significance levels is unclear. There are frequent trend changes from positive to negative, and therefore zero trends, which are nevertheless indicated as significant. So what is the test hypothesis?

Reply: We thank the reviewer for pointing at this problem. There was a typo in the plotting routine, which put wrong contours lines on top of the linear change plot. We apologize for not noticing that before. The significance areas are tested with the null hypothesis that there is no solar cycle or in the case of linear trends that a constant model would be sufficient. The corresponding p-values are very often much smaller than 0.1 and 0.05 corresponding to 90 and 95% significance levels. However, the drawn contour lines are computed from the error propagation as confidence level. We

tested whether the corresponding p-value contours match the confidence levels drawn from the error propagation. We have to note that p-values close to the thresholds, may should be considered with an own class.

We updated the figures with the linear changes and added the correct contours. Furthermore, we also modified the corresponding text in the manuscript.

Section 4.1, 1st paragraph: it may be worth to note that the seasonal cycle of the DT differs from the one presented in earlier papers like Portnyagin et al., 2004 or Jacobi 2012 in particular at lower heights, e.g., there is no or only weak spring/autumn enhancement. Earlier papers frequently used longer time windows for analysis. Differences then may be explained by strong day-to-day phase changes especially during summer. I would be interested in knowing whether this is the case.

Reply: We added some discussion how our results relate to the above mentioned previous studies. While analysing the data, we also investigated the effect of different epochs out of the complete time series and noted significant differences, which make a one to one comparison less conclusive. It appears that the time span that is analysed has a noticeable impact on the results. This becomes also very obvious by inspecting just the mean wind climatologies presented in Portnyagin et al., 2004. Although the same instruments are used, the mean wind climatologies already look slightly different.

The second aspect concerns the analysis method. The ASF (adaptive spectral filter) tends to be more robust to extract the tidal amplitudes for months, with a rapid change of the phase of the tides. The standard monthly mean vector summation, that is commonly used, my underestimate the amplitudes for such months. These phase drifts of the tides can exceed several hours over a month depending on season. This aspect requires further investigation. We add paragraph discussing this aspect.

We added to the manuscript: Comparing our climatologies to previous studies reveals some interesting differences. (Portnyagin et al., 2004; Jacobi, 2012) found a distinct maximum during the summer months and almost no tidal signature in winter at altitudes

from 92-98 km. Both studies did also not show the diurnal summer maximum below 82 km. These differences are partly explainable by the different length of the analyzed time series. Portnyagin et al. (2004) could only use a bit more than 1 year of data for the Scandinavian climatology. Jacobi (2012) compiled a climatology from 6 years during solar min conditions. However, the ASF decomposition of tides and mean winds considering the intermittent behaviour of the diurnal in amplitude and phase may also play a role.

The diurnal tidal phases are shown in Figure 10. The phases are referenced to a longitude of $13°$ east. The white contour line labels phase jumps and zero phases. The CMOR phases are shifted as if they would have been observed at the CMOR latitude, but at the above mentioned longitude in the European sector. The diurnal tidal phases show a distinct seasonal pattern and latitudinal differences. Throughout the year there are substantial changes in the phases at a given altitude, in particular at the polar latitudes during the winter months, the phases undergo phase drifts of several hours within a month.

Page 9, l 31, see also discussion and conclusion. Circular polarization means that u and v have the same amplitude and $90°$ phase difference. The latter is not shown.

Reply: Thanks for the remark. We agree with the referee that the term circular polarization is wrong for this case. After checking the phases, it turned out that there is not always a phase difference of $90°$.

Additional for the previous point we added Figures regarding phase information for the diurnal and semidiurnal tides and modified the text.

In the result part for the semidiurnal phase: Figure 14 shows the phase behaviour for the SDT. The seasonal pattern indicates an asymmetry and rapid phase change during the fall transition and the winter months. During the spring transition the phases also are altered, but less prominent compared to the fall and winter time. The phases also reflect the seasonal asymmetry similar than for the amplitudes of the SDT. Further, it

appears that latitudinal differences between Andenes and Juliusruh are small, whereas the phase differences to the CMOR latitude are much more significant. SDT phases also show continuous phase changes throughout the year. During the fall transition the phase changes within a month by more than 6 hours at all three latitudes. However, also the winter time is characterized by drifting SDT phases within a month.

In the discussions: The climatology of tidal phases for the DT and SDT point out that the tidal phases are not very stable at the MLT and more or less continuously changing throughout the course of the year. In particular, the rapid phase changes during the fall transition and the winter months (DJF) for the SDT are critical for many other analysis using long windows to determine tidal features. Typically, such long windows are used to separate the lunar tide from the SDT (Chau, 2015; Conte, 2017). However, already (Fuller-Rowell, 2016) pointed out that the phase stability is highly important in such an analysis. They found a lunar tide in model data (Whole Atmosphere Model - WAM) as a result of a drifting phase of the SW2 and TW3 tide during an SSW.

In the conclusions: The climatology of the tidal phases for the diurnal tide and the semidiurnal tide are not very stable. They change continuously through the year. Especially, during the fall transition and the winter occur phase changes for the semidiurnal tide.

< see Figure 1 in the supplements (Fig 9 in the manuscript) > Figure 9: Composites of the zonal (left) and meridional (right) diurnal phase information for Andenes (top), Juliusruh (middle) and CMOR (bottom).

< see Figure 2 in the supplements (Fig 13 in the manuscript) > Figure 13: Same as Figure 9, but for the semidiurnal tidal components.

Page 10, l 10. SSW are connected with mean wind changes at the timescale of days or few weeks, i.e. in the period range of PWs. So how do you distinguish between them? This also questions your statement on lines 11/12 that PWs are less strong during minor warmings.

Reply: We do not distinguish between the planetary wave activity and the SSW event as such. However, as planetary waves are the major driver for the development and the evolution of SSWs, we expect a clear connection between the occurrence of an SSW and the strength of the PW activity.

The planetary wave activity is estimated as residual of the daily mean winds and corresponding mean of a seasonal fit. The seasonal fit represents a model for each day of the year assuming undisturbed conditions, thus, the seasonal fitting provides a more robust estimate of the background (undistorted) wind than monthly averages. The seasonal fit provides a robust estimate of the background for all oscillations with periods shorter than 90 days (next harmonic of the seasonal fit). Planetary waves and SSW have characteristic time scales of days to weeks and hence should not bias our seasonal background fit.

We modified the text into: The planetary wave activity is estimated as residual between the daily mean winds, as obtained from the adaptive spectral filtering, and the seasonal fit shown in eq. (3). The seasonal fit provides a robust estimate of a background wind field for every day of the year and each wind component. The zonal and meridional wind residuals can be written as u' and v' and are considered as a good proxy of a planetary wave amplitude. However, this method does not allow to distinguish between a planetary wave like oscillation and a SSW event, which typically lasts 3-5 days at the MLT. However, Matsuno (1971) already pointed out that PWs play a major role in the evolution of SSWs. Figure 15 shows the PW energy. All three locations show striking enhancements during the winter, especially during years when a major sudden stratospheric warming (red arrow) takes place. During the years with a major sudden stratospheric warming event, the PW energy appears to be increased and takes values of up to 300 $m^2s^{-2}$ in the winter months. Minor sudden stratospheric warmings (green arrow) show also an increase of the PW energy, but usually weaker than during major sudden stratospheric warming. Even for the year 2016, where we found an exceptional circulation pattern at the MLT an enhancement of the PW energy is present Stober et

al. (2017), although this year did not show the evolution of a typical SSW (Matthias and Ern, 2018). For the rest of the year, the PW activity is comparatively low, with sparse enhancements observed at CMOR.

References: Matsuno, T.: A Dynamical Model of the Stratospheric Sudden Warming, Journal of the Atmospheric Sciences, 28, 1479–1494, https://doi.org/10.1175/1520-0469(1971)028<1479:ADMOTS>2.0.CO;2, 1971.

Matthias, V. and Ern, M.: On the origin of the mesospheric quasi-stationary planetary waves in the unusual Arctic winter 2015/16, Atmospheric Chemistry and Physics, 18, 4803–4815, https://doi.org/10.5194/acp-18-4803-2018, 2018.

Section 5: Earlier papers have shown that a possible solar cycle effect can be in phase, out of phase, or lagged with respect to the solar cycle. So here the phase should be provided to be able to compare with literature results.

Reply: We computed the phase information of our solar cycle fitting and found a very complex situation of all the different features we found above. Some of them seem to be correlated with the solar cycle other features are anticorrelated. Obviously, does the phase behaviour reflect a complex seasonal and altitude pattern that we also observe in the solar cycle amplitude plots.

We added a part into the manuscript which describes the phase information for the solar cycle and we added some figures into the appendix:

The phase information for the solar cycle can be found in the appendix. The phase is referenced to the year 2002. The yellowish to light orange colour indicate a zero phase shift compared to the reference year and corresponds to the maximum solar activity during solar cycle 23 (e.g., F10.7 or sunspot number). Phases that are outside the 90\% confidence interval are shaded and should not be treated as reliable due to the weak signal. The phase behaviour itself seems to be rather complex and depends on season and altitude. The phase pattern of mean wind shows also a strong latitude

dependence in both wind components. Only the winter season exhibits similarities in the response to the solar cycle of the sun with respect to the phase behaviour. There is a certain coherence of the phases between the latitudes in particular for the semidiurnal tide, which indicates a pronounced phase offset between September/October and December/January, which requires further investigation. However, the diurnal tide is basically only affected at the CMOR station and shows phases close to zero corresponding to a more or less direct response to the solar forcing. A more detailed discussion of the phase behaviour and the potential causes requires modelling and is beyond the scope of this paper.

We further added the following part into the conclusions:

- DT and SDT phases show a characteristic seasonal behaviour. The temporal evolution of the phases indicates continuous changes throughout the course of the year. SDT phases show rapid phase changes during the fall transition and at polar latitudes during the winter months (DJF). The mean phase behaviour as well as the continuous changes should be considered analyzing the lunar tides.

- Mean winds, DT and SDT show a seasonal dependent solar cycle effect and considerable different seasonal phase responses to the solar forcing. In particular, the SDT fall transition is characterized by an anticorrelation in September/October to the solar activity, whereas the winter months (DJF) seem to respond more directly to the solar forcing (e.g., F10.7 or sunspot number).

The Figures for the phase information of the winds and tides can be found in the appendix of the manuscript. The not significant areas are shaded.:

Beside the amplitude information of the solar cycle fitting (Figure 18 - 20), we also computed the phase information. These are shown for the wind component in Figure A1, for the diurnal tidal phase in Figure A2 and the semidiurnal tidal phase component in Figure A3. The shaded areas are not significant. It figured out to be a very complex situation of all the different features as described in Section 5. Some of them seem

to be correlated with the solar cycle and other anticorrelated. The phase behavior reflects a very complex time and altitude pattern that also be observed in the solar cycle amplitude plots.

< see Figure 3 in the supplements; Figure A1 in the manuscript > Figure A1: Phase information of the solar cycle fit for the wind component. The shaded areas are not significant.

< see Figure 4 in the supplements; Figure A2 in the manuscript > Figure A2. Same as Figure A1, but for the diurnal component.

< see Figure 5 in the supplements; Figure A3 in the manuscript > Figure A3. Same as Figure A1, but for the semidiurnal component.

Minor issues: General reply: We appreciate and adopted the comments regarding the technical corrections.

Abstract, l7: it is more common to handle the zonal and meridional wind components separately: ... the south and to the west. Reply: We reformulated "south-west" into "the south and to the west"

Abstract l 10: The influence of the 11-year solar cycle on the winds and tides is presented. Please provide the main result here. Reply: We added the following part to the abstract:

The influence of the 11-year solar cycle on the winds and tides is presented. The mean winds exhibit a significant amplitude response, for the zonal component below 82 km during summer and from November - December between 84 and 95 km at Andenes and CMOR. The SDT show a clear 11-year response at all locations, from October to November.

Page 2, l 27: with the exception of...do you mean "according to"? Reply: Thanks for the comment. We reformulated the sentence for a better understanding.

Page 2, l 34: waves restoring →wave ÌĄs restoring Reply: corrected

Page 3, l 12: I assume that T3 = 8 hrs, please add this information Reply: That's correct, T3 is the terdiurnal component. We added this information in the text, and also mentioned that the gravity wave residuum includes the terdiurnal part.

[. . .] Tn takes the values of 24 hours, 12 hours and 8 hours to determine the diurnal, semidiurnal and terdiurnal tide for each wind component. [. . .] The ASF provides a wave decomposition of our original observed time series into a daily mean wind, a diurnal and semidiurnal tide as well as a gravity wave residuum with an hourly resolution. Here, the gravity waves residuum also includes the terdiurnal tidal component.

Page 3, l 13: PW → GW Reply: The sentence is correct. For the estimation of the PWs we take for Tn one seasonal and the remaining residuals are defined as the GW residuum.

Page 3, l 16: nearly two weeks →ten days Reply: corrected

Page 3, l 28: Stober et al. 2012 does not deal with LTC Reply: Stober et al. (2012) showed the connection between changes in the neutral density and the prevailing zonal wind. Therefore it was mentioned here. Nevertheless, to avoid misunderstandings, we changed the citation into Stober et al. (2014), which sowed LTC of neutral density with respect to the solar cycle.

Page 4, l 2: Portnyagin et al., 2004 did not investigate LTC, but shows a long-term climatology Reply: we removed Portnyagin et al., 2004 from the list

Page 4, l 3: Lukianova et al., 2018 did not investigate LTC, but interannual variability, in particular SSW Reply: we removed Lukianova et al., 2018 here

Page 4, l 5: meridional wind trends by Jacobi et al., 2015 differ between seasons after 2005 Reply: we rewrote this part: From these studies, meteor radar wind observations show for the last decade seasonal dependent results for the mid-latitudes, with stronger eastward and southward directed tendencies during the autumn and winter,

and opposite tendencies during the spring e.g., Jacobi (2015).

Page 5, l 27: benefit is →benefit of Reply: corrected

Page6:, l 4: seasonal →annual Reply: corrected

Page 7, l 25: Jacobi et al. 2015 only showed data at ∼90 km, do you mean Jacobi, 2012? Reply: corrected

Page 7. l 35: 2006 →2007 Reply: corrected

Page 8, 2nd paragraph: The description is confusing. Please state clearly to which month and which altitude you refer to. Reply: we rephrased this part: At Andenes, an enhancement of the westward directed wind occurs during the begin of the year with values of up to 0.3 m/s/year, as well as for the summer in the area above the transition height. After the fall transition, a small enhancement of eastward winds is found, with values of up to 0.3 m/s/year for the complete observation height. The meridional wind for Andenes shows a pronounced southward directed wind long-term variability, with values of up to 0.5 m/s/year above ∼96 km for the winter and above ∼90 km for the summer. The LTC for Juliusruh is less significant, with changes which correspond to an eastward directed tendency during begin April and May and westward directed course below 90 km at June/July. Furthermore, an eastward enhancement below 90 km between September and November, with values of up to 0.5 m/s is found. The meridional component of Juliusruh shows tendencies towards south between January and April and an opposite tendency between May and December. At the location of CMOR, the strongest significant LTC occurs between April and August with an eastward acceleration of the zonal wind, enhancing the zonal jet above 90 km and weakening the westward jet below with values of up to 0.5 m/s/year. Meridional winds at CMOR show a southward long-term variability between 90 and 100 km at the beginning of the year and some northward accelerations in summer and fall.

Figure 12: again, there are zonal trends indicated as significant. Reply: We modified

the corresponding text a bit to explain the LTC of the SDT more clear.

The LTC for the semidiurnal tides is shown in Figure 12. All three MR exhibit slightly increasing SDT amplitudes around 90 km almost throughout the whole year. However, at Andenes and Juliusruh the significant changes emerge above 90 km during the early winter (November-December) showing a rather strong decrease of the SDT with amplitudes of 1 m/s/year. Additionally, a significant enhancement of the SDT occurs during the autumn transition showing an increase of up to 1 m/s/year. This behavior is not reflected at CMOR. There, it appears that the SDT amplitudes in November are further increasing in the zonal and meridional component. CMOR also exhibits a significant increase in the wintertime (December-February) of SDT amplitudes above 90 km.

Reference Portnyagin et al., 2004: provide doi: https://doi.org/10.5194/angeo-22-3395-2004 Reply: Thanks for the doi, we added it to the literature

Please also note the supplement to this comment:
https://www.ann-geophys-discuss.net/angeo-2019-51/angeo-2019-51-AC2-supplement.pdf
* * *
[Figure]

**Fig. 1.**

[Figure]

**Fig. 2.**

[Figure]

**Fig. 3.**

[Figure]

**Fig. 4.**

[Figure]

**Fig. 5.**

**Supplement:**

[revised manuscript text omitted]